# Aspirin mediates protection from diabetic kidney disease by inducing ferroptosis inhibition

**Ziyu Wu**[1,2], **Dan Li**[1], **Dingyuan Tian**[1], **Xuejun Liu**[1]\*, **Zhongming Wu**[1,3]\*

**1** NHC Key Laboratory of Hormones and Development, Tianjin Key Laboratory of Metabolic Diseases, Chu Hsien-I Memorial Hospital & Tianjin Institute of Endocrinology, Tianjin Medical University, Tianjin, China, **2** Department of Geriatric Medicine, Fujian Provincial Hospital, Fujian Provincial Center for Geriatrics, Fujian Provincial Key Laboratory of Geriatric Disease, The Provincial Clinical Medical College of Fujian Medical University, Fuzhou, Fujian, China, **3** Department of Endocrinology, Shandong Provincial Hospital Affiliated to Shandong First Medical University, Jinan, Shandong, China

\* xuejunliu2020@163.com (XL); wuzhongming@tmu.edu.cn (ZW)

**Data Availability Statement:** All relevant data are within the paper and its Supporting Information files.

**Funding:** This work was supported by the National Natural Science Foundation of China (32171339),

## Abstract

Diabetic kidney disease (DKD) progression can be predicted by abnormalities in the tubulointerstitial lining, and their treatment may be useful for preventing the disease. DKD is a progressive disease that contributes to renal tubular cell death, but its underlying mechanisms remain unclear. Ferroptosis is a novel term linked to lipid hydroperoxidation, and it plays an important role in the pathogenesis of DKD. Overexpression of cyclooxygenase-2 (COX2), an enzyme of the proximal tubule, causes cellular redox damage in DKD. It remains unknown whether COX2 exacerbates tubular damage by accelerating ferroptosis in the kidneys of diabetic mice. HK-2 cells cultured in high glucose exhibited ferroptosis, which was inhibited by ferroptosis inhibitors. Additionally, alterations in the sensors of ferroptosis metabolism, such as glutathione peroxidase 4 (GPX4) activity, lipid hydroperoxidation, reduced glutathione (GSH) levels and changes in mitochondrial morphology, were observed in high glucose-cultured HK-2 cells. Diabetic mice manifested tubular injury and deranged renal physiological indices, which were mitigated by ferrostatin-1 (Fer-1). Importantly, these perturbations were ameliorated by downregulating COX2. In addition, the increased COX2 was observed to be elevated in the daibetic kindney. To explore the relevance of COX2 to ferroptosis, HK-2 cells that knocked down from COX2 exhibited decreased ferroptosis sensitivity under high glucose conditions. In RSL-3-treated HK-2 cells, ferroptosis was improved by downregulating COX2 by treatment with aspirin, which was confirmed in high glucose-cultured HK-2 cells. Furthermore, the ferroptosis changes were also suppressed by decreasing COX2 in diabetic mice treated with aspirin, which retarded DKD progression. In conclusion, our results demonstrated that ferroptosis in renal tubular cells contributes to DKD development and that diabetes-related ferroptosis was inhibited through the downregulation of COX2 by aspirin, thus retarding the progression of DKD. Our findings support a renoprotective mechanism by which aspirin inhibits COX2 activation, identify COX2 as a potential target of ferroptosis, and establish that ferroptosis in renal tubular cells is an integral process in the pathogenesis of DKD regulated by COX2 expression profiles.

the Key Program of Natural Science Foundation of Tianjin (19JCZDJC36900), Fujian Provincial health technology project (2019-1-22) and the Science and Technology Self-financing Project of Tianjin Municipal Commission of Health (ZC20143).

**Competing interests:** The authors have declared that no competing interests exist.

## 1. Introduction

Diabetic kidney disease (DKD) is one of the leading causes of end-stage renal disease (ESRD) and is epidemic around the world. It is estimated that the total diabetes prevalence will reach 33% of the US adult population by 2050 [1]. DKD is characterized by proteinuria and is generally attributed to glomerular abnormalities [2]. It has been reported that blocking the renin-angiotensin system can retard the progression of kidney disease and supplies partial renoprotection. However, approximately 30% of patients with diabetes on angiotensin receptor blockers (ARBs) or angiotensin-converting enzyme inhibitors (ACEi) progress to ESRD [3], and there is a large unmet need to develop more effective treatments for DKD patients. Although the exact pathogenesis of DKD is still unresolved, ferroptosis-related renal tubular injury is suggested to be associated with DKD. Therefore, in addition to developing additional therapies to lessen proteinuria, protecting proximal tubular epithelial cells (PTECs) from ferroptosis may provide an approach to ameliorate outcomes in patients with DKD.

Tubular cell death is the major early step in the pathogenesis of DKD and is accompanied by inflammation induced by damage-associated molecular patterns and chemokines of dead and dying cells. Altered tubular cell death signals are also implicated during DKD. The best parameter of kidney disease functional progression is tubulointerstitial disease [4]. Tubular abnormalities may precede the onset of early glomerulopathy in DKD [5]. The pathogenesis of DKD, which may initiate tubular epithelial degeneration and injury, is still poorly understood. Recently, the renoprotective effects of sodium-glucose cotransporter-2 (SGLT2) inhibitors have confirmed PTECs as a target of therapy [6]. Many factors, including ischemia, hypoxia, mechanical stretch, ROS and inflammatory cytokines, can cause tubular cell death [7–10]. However, the relationship between fibrosis and tubular cell death during the progression of DKD is still unclear. Further exploring the pathogenesis of tubulointerstitial disease will provide new treatment strategies for DKD.

Ferroptosis is a novel nonapoptotic cell death that was discovered to be iron-dependent in 2012 and induced by redox imbalance redox homeostasis, which is characterized by detoxification of lipid oxidation products and free radicals [11]. It is caused by the loss of antioxidant capacity, redox-active iron accumulation and intracellular lipid hydroperoxide accumulation [12]. The pathogenesis of ferroptosis can be inhibited by glutathione peroxidase 4 (GPX4) [13]. The products of lipid peroxidation involve reactive aldehydes (e.g., 4-hydroxynonenal (4HNE) and malondialdehyde (MDA)), which increase during ferroptosis [14]. In the presence of ferroptosis, the process of System Xc- was disrupted, depleting sufficient GSH production to prevent GPX4 from exerting its antioxidant capacity [15,16]. Because renal tubular cells are enriched in polyunsaturated fatty acids (PUFAs) [17], the kidney is susceptible to ferroptosis induced by lipid hydroperoxides. It has been demonstrated that ferroptosis pathogenesis is the cause of folic acid-induced AKI and IRI-induced ATN [18,19], but its role in DKD remains to be investigated. These results suggest that inhibiting ferroptosis in injured PTECs may become a new treatment for kidney diseases.

Cyclooxygenase-2 (COX2) is a specific enzyme of the renal proximal tubule that was recognized several years ago [20]. COX2 has recently attracted greater attention because COX2 expression occurs early in the disease process by physiological stimuli or inflammatory cytokines [21–24]. It catabolizes arachidonic acid to various prostanoids, including thromboxane and prostaglandins, and it plays an important role in renal tubular damage. COX2 is a cyclooxygenase superfamily protein that is an inducible enzyme at low levels in most tissues and is significantly increased in the proximal tubule of the injured kidney [25,26]. Moreover, upregulated COX2 expression also induces increased production of ROS and aggravates renal

tubular damage in many pathological conditions [27]. However, its role in the pathogenesis of ferroptosis remains to be explored.

Aspirin is a nonspecific inhibitor of COX2 activation. Various recent clinical trials have indicated that aspirin lessens the risk of cardiovascular disease and improves renal disease progression in diabetes [28,29]. Notably, a clinical observational study demonstrated that aspirin markedly reduced proteinuria in DKD patients [30]. While the role of COX2 expression in aspirin-mediated cardioprotection is still controversial, its role in renoprotection remains unclear. Thus, showing the mechanism of aspirin-mediated renoprotection may supply a novel target for the pathogenesis of DKD. Studies have demonstrated that aspirin can protect cells from the production of thromboxane and reduce platelet hypersensitivity to inhibit the progression of diabetic complications, but there is still debate about its exact mechanism [31]. In addition, inhibited COX2 expression in tissues may be involved in aspirin-mediated renoprotection. However, the mechanism by which COX2 regulates ferroptosis has not been elucidated. We now show that aspirin can markedly lessen proteinuria and retard diabetic kidney disease progression by inhibiting the protein COX2. Because aspirin is a nonspecific inhibitor of COX2 activation, this research aimed to explore the underlying mechanisms of aspirin in the treatment of tubulointerstitial injury in diabetic mice. Therefore, we hypothesized that inhibiting ferroptosis prevents PTEC damage by downregulating COX2 expression signaling, and this mechanism may illustrate how aspirin mediates renoprotection.

Due to the presence of COX2 changes and ferroptosis markers on the renal tubules, we explored the COX2 pathobiology and ferroptosis of tubulointerstitial injury in diabetic kidney disease. Our research shows that inhibiting ferroptosis has renoprotective properties by downregulating COX2 signaling in PTECs, which illustrates the mechanism of aspirin-mediated renoprotection in DKD. This study is expected to supply novel avenues for treating DKD.

## 2. Material and methods

### 2.1. Cell culture studies

In a humidified environment, human proximal tubular epithelial cell line (HK-2) cells were cultured with 5% CO2 at 37°C. They were maintained in Dulbecco's Modified Eagle Media (DMEM; Gibco, Carlsbad, CA, USA), consisting of 10% fetal bovine serum (FBS; Biological Industries, Cromwell, CT, USA) and 100 U/mL streptomycin and penicillin (Solarbio, Beijing, China) in Ctrl group. In HG group, cells were maintained in DMEM containing 30 mmol/L glucose supplemented with100 U/mL streptomycin and penicillin and 10% FBS. In Mannitol group, cells were maintained in DMEM with 5.5 mmol/L glucose, 24.5 mmol/L mannitol, 100 U/mL streptomycin and penicillin and 10% FBS. Mannitol group was applied to control cell osmolality. Ferrostatin-1 (Fer-1, Selleck, Houston, TX, USA, 400nM) and Aspirin (Solarbio, Beijing, China, 400uM) were used to treat the cells. In additional, COX2-siRNA (Sangon Bitotech, Cat. SR310776, 50 nM) in cells transfected with Lipofectamine 2000 regent (*US Everbright Inc*, Cat.L7003) was used to knockdown studies. In Fer-1 group, AS group and DMSO group, cells were respectively cultured in 30 mmol/L glucose medium supplemented with 400nM Fer-1, 400 μM Aspirin and 0.1% DMSO for 48 h. In Si-COX2 group, cells were respectively treated with COX2-siRNA transfections and cultured in 5.5 mmol/L glucose or 30 mmol/L glucose medium for 48 h. In RSL-3 group, RSL-3 + AS group and DMSO group, cells were respectively cultured in 5.5 mmol/L glucose medium supplemented with 100nM RSL-3, 100nM RSL-3 plus 400 μM aspirin and 0.1% DMSO for 48 h. And in Si-COX2 + RSL-3 group and Si-COX2 + DMSO group, cells were treated with COX2-siRNA transfections and maintained in 5.5 mmol/L glucose medium supplemented with 100nM RSL-3 and 0.1% DMSO for 48 h.

## 2.2. Animal model system

All animal research experimental protocols were conducted adhered to the National Institutes of Health Guide for the Care and Use of Laboratory Animals. The ethical committee of Tianjin Medical University approved this study. These mice were on eight weeks old male DBA/2J background ($n = 36$, HFK Bioscience, Beijing, China). They were randomized one of the six groups: control normal mice group (NC); diabetic mice group (DM); diabetic mice group (Fer-1),who intraperitoneal injected Fer-1 (Selleck, Houston, TX, USA); diabetic mice group (vehicle-P),who intraperitoneal injected 1% dimethyl sulfoxide (DMSO); diabetic mice group (As),who intragastric administrated Aspirin (Solarbio, Beijing, China); diabetic mice group (vehicle-G),who intragastric administrated 0.5% sodium carboxymethyl cellulose (Na-CMC; Solarbio, Beijing, China). Diabetes models were induced with 5 consecutive days of a single intraperitoneal injection of streptozotocin 40 mg/kg (dissolved in 0.1 M citrate buffer, pH 4.5; SigmaAldrich, St Louis, MO, USA). Control mice only was injected the same volume of citrate buffer. In the Fer-1 or vehicle-P groups, the diabetic mice were treated respectively with Fer-1 (2.5 µmol/kg, dissolved in 1% DMSO) or 1% DMSO during the duration of treatment for 12-week every day. And in the AS and vehicle-G groups, the diabetic mice were treated respectively with aspirin (50 mg/kg, dissolved in 0.5% Na-CMC) or 0.5% Na-CMC for 12-week every day. Plasma glucose were measured from tail vein blood. Mice with random plasma glucose concentrations >300 mg/dL (16.7 mmol/L) for 3 days after the last injection of streptozotocin were used for further study. All mice (3–5 mice per cage) were exposed to standard laboratory conditions (12 h on/off; lights on at 7:00 a.m.) and were housed at 10% humidity with an ambient temperature (22–24°C) ad free access to food and water. Mice body weight and blood glucose levels were measured at week 0, 2, 4, 8 and 12. Mice were respectively housed in metabolic cages for 24 h, and urine samples were gathered and urine output was recorded at week 0 and 12. Urinary creatinine concentrations and Urinary albumin were assayed by the Mouse Cr ELISA Kit (Fankewei, Shanghai, China) and Mouse uALb ELISA Kit (Fankewei, Shanghai, China). Then urinary albumin/creatinine ratio (ACR) and urinary albumin were calculated. All mice were euthanized with 60 mg/ml of pentobarbital sodium at the end of the experimental period. Kidneys were harvested at the time of sacrifice and stored for examination.

## 2.3. Morphological evaluation and tubular damage scoring

Paraffin-embedded tissue sections (4 µm thick) were heated and deparaffinized with xylene. The sections were rehydrated by gradually decreasing ethanol concentrations. Following the slides were stained sequentially by Hematoxylin (3 min) and Eosin (30 sec). After distilled water rinsed a short, the slides were dehydrated by ethanol, cleared in xylene, mounted with covers lip and assessed. The slides were mounted with neutral balsam at last. The degree of renal injury, including renal tubular injury, mesangial matrix expansion and interstitial fibrosis was analyzed by Image-Pro Plus 6.0 software. HE-stained sections were applied to evaluate disruption of tubular epithelium and dilation of tubule. For each mice, two investigators assessed100 tubules from 10 different views. Seven parameters were graded as follows: brush border loss(0 or 1),flattening of tubular epithelial cell (0 or 1), vacuolization of cytoplasm (0 or 1), bleb formation of cell membrane (0 or 2), obstruction of tubular lumen (0 or 2), edema of interstitium (0 or 1), and cell necrosis (0 or 2). Cumulative values in the range 0 (unaffected) - 10 (severely damaged) were considered an indication of tubular damage extent. For Periodic acid-Schiff's (PAS), dehydrated slides were incubated with Periodic Acid Solution for 10 min. Then distilled water washed and Schiff Solution immersed in the slides for 30 min. Following distilled water rinsing, Hematoxylin stained the slides for 3 min. After distilled water washing

again, Bluing Reagent TM was used for 30 sec, then slides were rewashed, dehydrated, cleared in xylene and mounted with covers lip. The slides were assessed for the mesangial matrix expansion by optical microscopy. Measuring the relative number of pixels (pink or red areas) divided by the total area of the glomerulus to evaluate semi-quantitatively Mesangial matrix expansion. Randomly selected 20 stained glomeruli each slide (3 mice/group) were analyzed. The extent of renal fibrotic areas were assessedby Masson staining with estimating the relative number of pixels (blue).

## 2.4. Immunohistochemical (IHC) studies

Paraffin tissue sections of kidneys(4 μm thick) were deparaffinized and rehydrated. Following heat-induced epitope retrieval, phosphate buffer saline (PBS) washed the sections for 3 times. To block the background staining, all sections were incubated with 3% BSA in PBS at room temperature for 1 hour. The sections are immersed respectively in COX2 antibody (A1253; ABclonal;1:100),GPX4 antibody (A11243;ABclonal,1:100), FTH-1 antibody (DF6278; Affinity;1:100), and SLC7A11 antibody(DF12509;Affinity;1:100) overnight at 4˚C and washed with PBS. HRP conjugated secondary antibody (Vector Laboratories) were incubated in the sections at room temperature for 1 hour. Then rinsing with PBS, to develop the peroxidase reaction, productdiamino-benzidine (DAB) solution was used for 20–35 seconds. All sections were following washed immediately with distilled water and counter stained with Hematoxylin, dehydrated in graded series of ethanols, mounted with covers lip and assessed by optical microscopy.

## 2.5. Immunofluorescence (IMF) studies

Variously treated HK-2 cells were fixed in 4% paraformaldehyde. Then they were permeabilized by 0.25% Triton X-100 in PBS buffer. 2% BSA in PBST was appplied to block the background staining. After primary antibodies were used to incubated with the cells overnight at 4˚C: GPX4 antibody (A11243; ABclonal,1:100). Then the cells were rinsed with PBS and the secondary antibodies goat anti-rabbit IgG were incubated for 1 h and counter stained with 4,6-diamidino-2-phenylindole (DAPI; Solarbio, Beijing, China) at the room temperature. Subsequently, being rinsed with PBS for 3 times, the cells were then mounted with covers lip and observed by fluorescence microscopy.

## 2.6. Transmission electron microscopy of cells

HK-2 cells pellets were fixed with 2.5% glutaraldehyde (Alfa Aesar, Ward Hill, MA, USA) in 0.1 M phosphate buffer saline (pH 7.4) at 4˚C for 3 h. After incubated with 1% aqueous osmium tetraoxide and 0.1 M phosphate buffer saline (pH 7.4) at room temperature for 2 h, dehydrated in graded series of ethanols (50%–100%), embedded in epoxy resin monomer and cured at 60˚C for48 h. Ultra-thin sections (50 nm) were stained with uranyl acetate and lead citrate. Two samples were prepared for each group, and each sample were randomly selected five fields of view and observed by transmission electron microscopy (HT7700-SS; HITACHI, Tokyo, Japan).

## 2.7. Western blotting analysis

Immuno-blotting were applied to assess various samples protein expression. The tissues were diced into 1 mm$^3$ and homogenized in RIPA lysis buffer (Solarbio, Beijing, China). Furthermore the HK-2 cells were lysed with RIPA lysis buffer. Centrifuge the homogenate at 10,000×g for 5 minutes and collect the supernatant. After each of the samples were adjusted the concentration (100 μg/100 μl), the SDS loading buffer were mixed with equal amounts of protein

(20 μg), heated for 10 min at 100˚C, followed by centrifugation and vortexing to remove undissolved fragment, ice-cooled and subjected to 7.5,10 or 12.5% SDS-PAGE. After the proteins was separated on SDS-PAGE, they were transferred to PVDF membranes by electro-blotting procedures. Following the membranes were blocked with 0.1% tris buffered saline tween that contained 5% skimmed milk at room temperature for 1 h, members were individually incubated with each primary antibodies overnight at 4˚C: GPX4 antibody (A11243;ABclonal,1:1000), FTH-1 antibody (DF6278; Affinity;1:1000), and SLC7A11 antibody(DF12509; Affinity;1:1000), TFR-1 antibody (DF214039;Affinity; 1:1,000), COX2 antibody (A1253; ABclonal;1:1000), and GAPDH antibody (60004-1-Ig;Proteintech;1:5,000). They were rinsed with TBST buffer and incubated with the normal isotype-matched horseradish peroxide-labeled IgG (1:1000) for 60 min at room temperature. Following another wash with TBST, the ECL chemiluminescence reagent (Amersham Biosciences) was applied to visualize the immunoblots, and densitometry was performed by Image J software v1.49.

## 2.8. Quantitative real-time PCR (qRT-PCR)

The gene expression in various samples were evaluated by Real-time PCR. HK-2 cells and 10 mg of kidney tissue were used to isolate RNA by using the HP Total RNA Kit (Omega Bio-Tek, Norcross, GA, USA). The RNA was reverse transcribed into cDNA using M-MuLV First Strand cDNA Synthesis Kit (Sangon Biotech, Shanghai, China). For quantitative PCR (qPCR), reaction mixture, containing 1 μmol/L of forward and reverse primer,100 ng cDNA, 2 μl of nuclease-free water and 1 × Fast SYBR Green Master Mix, in a total volume of 20 μl was prepared. The C1000 Touch Thermocycler CFX96 Real-Time System (Bio-Rad, Hercules, CA, USA) was performed to PCR. The mRNA levels of different samples was calculate byRelative CT values compared with GAPDH. The primers used in the research were listed in S1 Table.

## 2.9. Transfections

The cells were transfected with 50 nM of siRNA (Sangon Bitotech, Shanghai, China) individually against COX2(Si-COX2) and control siRNA (Si-Ctrl), performing with lipofectamine RNAimax (*US Everbright Inc*, Cat.L7003) to generate COX2 knockdown studies. Target sequences for preparing siRNAs of human COX2 are listed in S1 Table. 250 μL of Opti-MEM serum-free media (Gibco, Carlsbad, CA, USA)was combined with 5 μL of 20 μM siRNAs in one tube. 250 μL of Opti-Mem media was combined with 5 μL of lipofectamine 2000 in another tube. They were equilibrated for 5 min at regular temperature. Two tubes were gently mixed and incubated for 20 min at 37˚C, transferred into 6-well plate combined with 1.5 mL of Opti-MEM serum-free media. Performing Quantitative real-time PCR to confirm the siRNA induced gene silencing.

## 2.10. Assessment of intracellular ROS in HK-2 cells

$2', 7'$ -dichlorofluorescin diacetate (DCHF-DA, Solarbio, Beijing, China) was applied to evaluate intracellular ROS level. The cells were stained with DCHF-DA at 37˚C for 30 min. we rinsed excess dye out with phosphate buffer saline in the dark. Following fluorescence intensity of cells were observed by fluorescence microscopy with excitation at 488 nm and emission at 525 nm.

## 2.11. GSH levels of kidney tissues and cells assay

About 20mg kidney tissues or ~2 x 106 cells with 90–95% confluency per culture dish (55 cm2) were applied to evaluate the GSH levels. The tissues were homogenized with 200 μl GSH

Extraction Buffer. Besides the cells were lysed in 50 μl GSH Extraction Buffer. Then the samples were shortly centrifuged and vortexed at 8,000 x $g$ for 10 min. 20 μl upernatants or standards samples were added into the respective wells of 96 well-plate. GSH was detected by Micro Reduced Glutathione GSH Assay Kit (Solarbio,Beijing, China).When the reaction period was over, colorimetric reading were made by using a Microplate Reader (BIO-RAD) at the wavelength of 412 nm. At last, the GSH levels of the samples was calculated by comparing the optical density of the samples to the standard curve.

### 2.12. Iron assay

The iron levels were detected by the Iron Assay Kit (BioAssay, Hayward, CA, USA), transfer the working reagents and collected supernatant to a 96-well plate. Flow cytometric reading was made at the wavelength of 590 nm.

### 2.13. CCK8 assay

According to the manufacturer's instructions, the viability of cell was detected by applying the Cell Counting Kit-8 (Dojindo, Kumamoto, Japan). HK-2 cells were seeded at a density of 3000 cells per well into the 96-well plate (~3,000 cells/well). The cells were administrated with different concentrations of the compounds to indicate times. After treatment, the growth medium was removed and cells were cultured with 10 μL work reagent at each well and incubated at 37˚C for 2 h. A microplate reader (Synergy HT, Bio-Tek, United States) detected the absorbance at 450 nm. Orbitally shake the 96-well plate for 10 min to completely dissolve the formazan. Then measure the absorbance at 490 nm by a microplate reader (Synergy HT, Bio-Tek, United States).

### 2.14. MDA and 4-HNE assay

10 mg tissues of kidney were sliced and rinsed with cold saline in each group. Immediately homogenize the tissue by shaking the homogenizer with cold saline. Otherwise, the ultrasonic cell disrupter homogenized the cells. To detect the MDA concentrations of tissues and cells,the supernatant was collected after centrifugation for 10 min. The concentrations of MDA were measured by Micro Malondialdehyde Assay Kit (Solarbio, Beijing, China). Then transfer the working reagent and collect the supernatant into a 96-well plate. To calculate the MDA concentration,the optical density was detected at the wavelength of 600, 532 and 450 nm. After tissues of kidney were homogenized, the supernatant was collected after centrifugation. In addition, the ultrasonic cell disrupter homogenized the cells. 4-HNE concentrations was measured by 4-HNE ELISA Kit (Fankewei, Shanghai, China). Microplate reader detected the optical density of the supernatant at the wavelength of 450 nm. And 4-HNE concentrations of the supernatant were then obtained by comparing the optical density of the samples to the standard curve.

### 2.15. Statistical analysis

Student's $t$-test was used for the difference analysis of two groups, and one-way ANOVA analysis was performed for multiple group comparison. $p$ less than 0.05 was considered significant. Data were shown as mean ±SD. GraphPad Prism 8.0 software (GraphPad, La Jolla, CA, USA) and Microsoft Excel 2013 were used for all calculations.

## 3. Results

### 3.1 Ferroptosis is an essential part of high glucose-cultured HK-2 cells

Under treatment with various media with different glucose concentrations (5.5, 10, 15, 20, 25, 30, 35, 40, 45 and 50 mmol/L) for 48 h, the viability of the cells was measured by CCK8 assay.

HK-2 cells cultivated in medium containing 30 mmol/L glucose had a higher viability than those cultivated in medium containing 5.5 mmol/L glucose (Fig 1A). DCHF-DA staining (2',7'-dichlorofluorescein diacetate) showed that high glucose-cultured HK-2 cells generated more ROS than the control cells. The Ctrl group cells exhibited a minimal degree of fluorescence, and cells cultured in high glucose alone generated the greatest background fluorescence. The generation of ROS was significantly reduced by cotreatment with 400 nm Fer-1 (Fig 1B). To highlight the role of ferroptosis, our study observed the dynamics of intracellular iron, i.e., ferrum and antioxidant stress-related protein, i.e., solute carrier family 7 member (SLC7A11), glutathione peroxidase 4 (GPX4) with iron metabolism-related protein, i.e., heavy chain of ferritin (FTH1), transferrin receptor 1 (TFR-1). Western blot analysis was applied to indicate that COX2 and TFR-1 protein expression was increased, and the expression of SLC7A11, GPX4 and FTH-1 were decreased in HG group cells compared with the Ctrl group (Fig 1C). Additionally, quantitative real-time PCR results revealed that the mRNA expression levels of GPX4, SLC7A11, FTH-1 and TFR-1 and the corresponding proteins were changed in parallel. Thus, the changes in ferroptosis-related genes were reversed in the Fer-1 group (Fig 1D–1G). The intracellular GSH levels were assayed by a Micro Reduced Glutathione GSH Assay Kit. The research demonstrated that GSH was also clearly decreased in HG group cells (Fig 1H). Simultaneously, the data showed the end products of lipid peroxidation; a micro malondialdehyde assay kit was used to assess the level of malondialdehyde (MDA), and the production of MDA was increased by 50% (Fig 1I) compared with that in the Ctrl group. Another products of lipid peroxidation, 4-HNE(Fig 1J) was dramatically elevated in HG group cells. Furthermore, the concentration of iron (Fig 1K) was increased by 80% in HG group cells compared to Ctrl group cells. However, Fer-1 significantly inhibited the production of MDA and enhanced the intracellular GSH levels and iron concentration in high glucose-cultured HK-2 cells. Thus, the above mentioned findings demonstrated that the pathogenesis of ferroptosis was caused by high glucose in vitro, and we further explored its biology in vivo.

## 3.2 Ferroptosis inhibition attenuates the renal pathological changes of kidney fibrosis in diabetic kidney disease

We investigated whether ferroptosis inhibition can have the renoprotective effects of Fer-1 treatments in DKD. To evaluate the effects of Fer-1 on an experimental model of streptozotocin (STZ)-induced type 1 diabetes, five consecutive injections of low-dose streptozotocin (STZ) (40 mg/kg/day intraperitoneally (IP)) in 8-week-old male DBA/2J mice resulted in the development of diabetes (Fig 2A). To elucidate the pathogenesis of ferroptosis in diabetic kidney disease, diabetic mice were treated with Fer-1 for 12 weeks. All mice were monitored after STZ injection for 3 months before sacrifice. The body weight was gently increased in the NC group, while it was progressively decreased in the Fer-1, DM and vehicle-P groups (Fig 2B). Meanwhile, random blood glucose progressively increased and attained 25 mmol/L in the Fer-1, DM and vehicle-P groups at week 12, but it was retained at approximately 7 mmol/L in the NC group (Fig 2C). To evaluate renal function, the renal dysfunction of diabetic mice was defined by high levels of the 24-h urine volume (Fig 2D), urinary albuminuria (Fig 2E) and ACRs (Fig 2F) in the Fer-1, DM and Vehicle-P groups. In particular, compared with the normal mice, the 24-h urine volume and urinary albuminuria were elevated by 30 and 100 times in the DM group. Diabetic mice had significantly higher kidney weights than NC mice (Fig 2G). These four quotas of the mice in the Fer-1 group were higher than those of the NC group, while they were clearly lower than those indicators of the DM group. HE staining showed changes in renal morphology, including tubular epithelial degeneration and brush border loss, with dilatation of the tubular lumen in diabetic mice. These tubular injuries were clearly

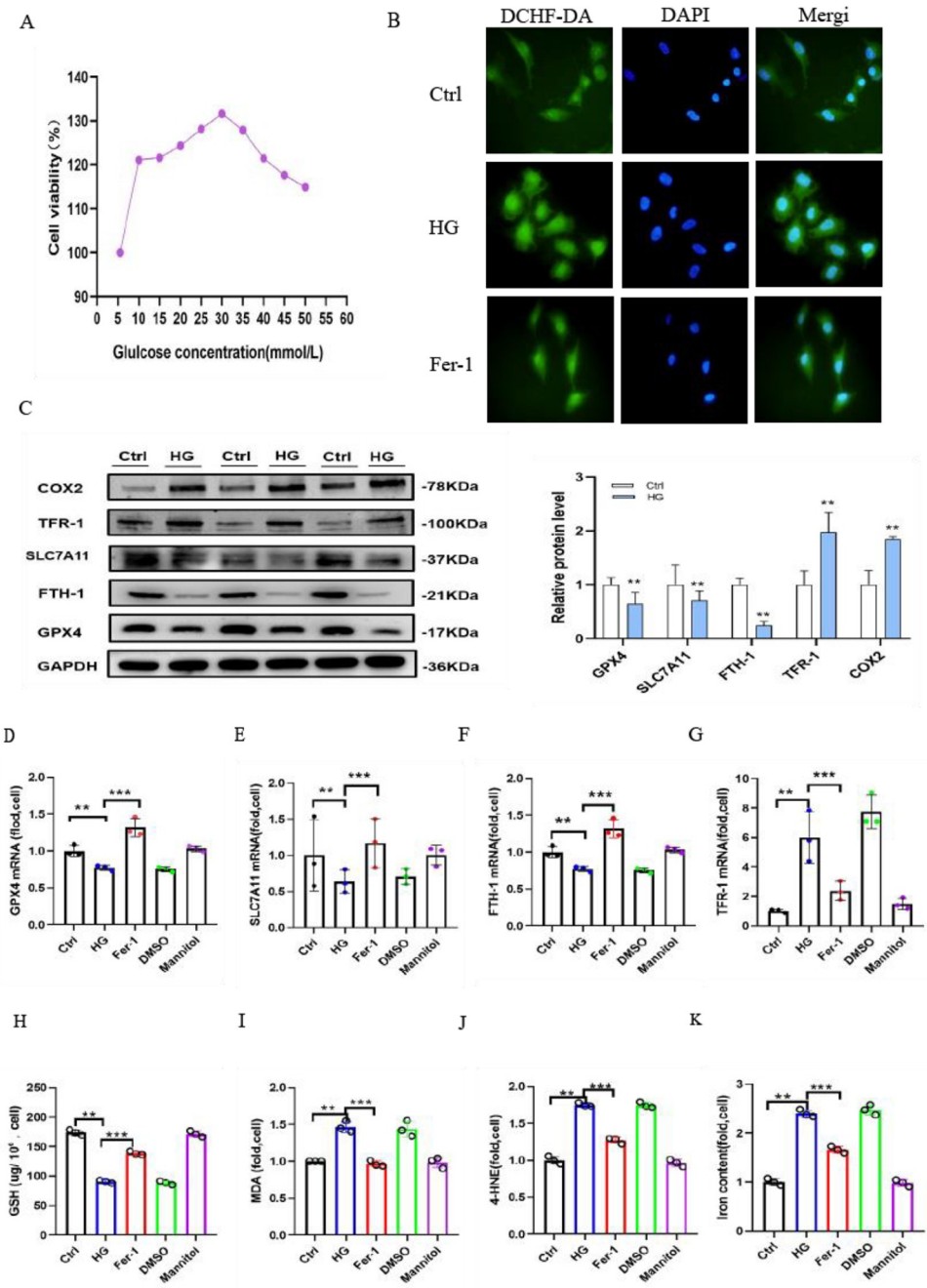

**Fig 1. Ferroptosis is a critical component of high glucose-cultured HK-2 cells.** (A) CCK8 was used to detect the viability of HK-2 cells cultured in different concentrations of glucose medium for 48 h. (B) DCHF-DA staining revealed the generation of ROS (green) in high glucose-cultured HK-2 cells compared to the control cells, which was partially reduced by Fer-1 treatment for 48 h. (C) Western blot analysis of GPX4, SLC7A11, FTH-1, COX2 and TFR-1 protein expression in cells in the Ctrl and HG groups. (D-G) GPX4 (D), SLC7A11 (E), FTH-1 (F) and TFR-1 (G) mRNA levels in cells in the Ctrl, HG, Fer-1, DMSO and mannitol groups.(H-K) GSH (H), MDA (I),4-HNE(J) and iron concentrations (K) of cells in the Ctrl, HG, Fer-1, DMSO and mannitol groups. Data are expressed as the mean ± SD. $^{**}p < 0.05$ *vs*. Ctrl group; $^{***}p < 0.05$ *vs*. HG group.

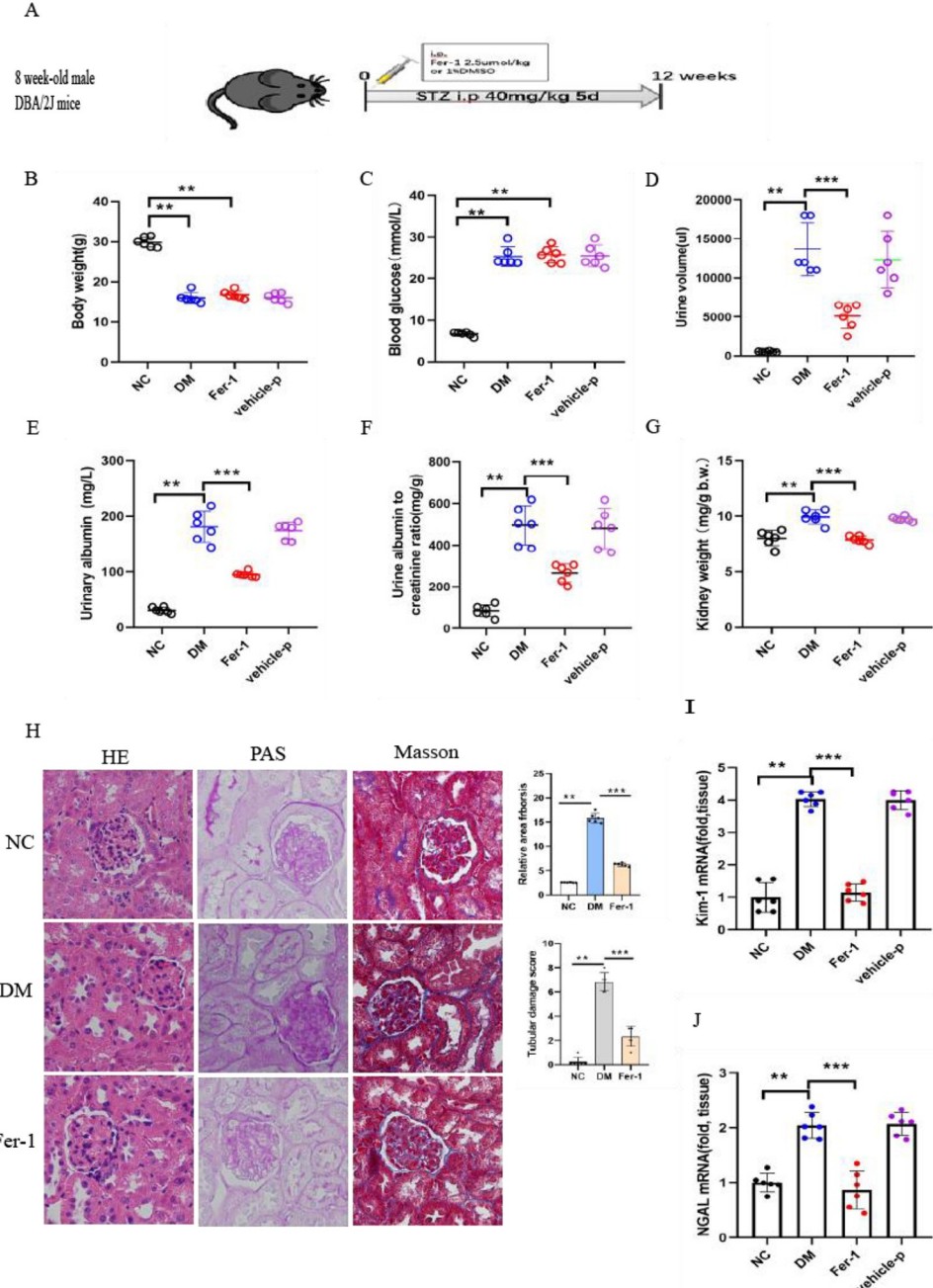

**Fig 2. Inhibition of ferroptosis by Fer-1 ameliorated the pathological changes of kidney fibrosis in diabetic kidney disease.** (A) Schematic diagram. Five doses of STZ (40 mg/kg/day IP) were injected to show the induction of diabetes in DM and Fer-1 mice. (B-G) Body weight (B), blood glucose (C), 24-h urine volume (D), urinary albuminuria (E), albumin-to-creatinine ratio ACR (F) and kidney weight/body weight (G) were monitored in the NC, Fer-1, DM and vehicle-P groups. (H) HE staining, PAS staining and Masson staining of kidneys in nondiabetic and diabetic control Fer-1 mice. The relative area of fibrosis (%) and relative collagen (%) were assessed by the ImageJ program. (I-J) The increased KIM-1 and NGAL mRNA expression was relieved by Fer-1 in diabetic mice. Data are the mean ± SEM. $^{**}p < 0.05$ *vs.* NC group; $^{***}p < 0.05$ *vs.* DM group.

attenuated by Fer-1 treatment (Fig 2H). PAS staining revealed clear mesangial expansion and a loss of brush borders with more dilated cortical proximal tubules and numerous exfoliated tubular cells compared with the Ctrl group (Fig 2H). Masson staining revealed basement membrane atrophy, cortical proximal tubule thickening, and relative collagen deposition in the lumen of the DM group. Moreover, this renal damage was remarkably reversed in diabetic mice treated with Fer-1 (Fig 2H). Of note, the scores of tubular damage and KIM-1 and NGAL mRNA were increased in the kidneys of diabetic mice. These changes were alleviated by Fer-1 (Fig 2I and 2J), demonstrating that notable renal functional deterioration induced by high glucose is attenuated by the inhibition of ferroptosis.

## 3.3 Renal ferroptosis and COX2 expression are observed in diabetic mice

It is believed that both SLC7A11 and GPX4 are involved in antioxidative stress. In this research, SLC7A11 and GPX4 were expressed at lower levels in the renal proximal tubules of diabetic mice by immunohistochemical staining and semiquantification analyses (Fig 3A). The protein and mRNA expression levels of SLC7A11 and GPX4 in the DM group were clearly decreased compared with those in the NC group (Fig 3B–3D). This study also showed that the iron metabolism-related genes TFR-1 and FTH-1 were changed in diabetic mice. FTH-1 mRNA expression was dramatically decreased, and TFR-1 and PTGS2 were significantly increased in diabetic mice (Fig 3E–3G). Meanwhile, immunohistochemical staining showed that FTH-1 staining was weaker and COX2 staining was stronger in renal sections of the DM group than in the control group (Fig 3A), which was consistent with the corresponding protein expression changes (Fig 3B). The obtained data showed that GSH was also clearly decreased in the kidneys of diabetic mice (Fig 3H), indicating that the capacity for peroxidation repair was significantly reduced. Moreover, the obtained results showed that MDA (Fig 3I) and 4-HNE (Fig 3J), which are the end products of lipid peroxidation, were dramatically elevated in diabetic mice. Furthermore, the iron concentration in the kidneys of diabetic mice was approximately four times higher than that of normal mice, demonstrating iron overload in the kidneys of the DM group (Fig 3K). The decreased GSH levels were improved, and the increased kidney iron content and MDA and 4-HNE levels were blocked by Fer-1 treatment in diabetic mice. Interestingly, these results illustrated that the abovementioned ferroptosis-related changes were significantly reversed by the ferroptosis inhibitor Fer-1 in diabetic mice.

## 3.4 High glucose promotes ROS generation, and COX2 gene disruption alleviates high glucose-induced ferroptosis

We examined the effect of aspirin-mediated renoprotection in high glucose-cultured HK-2 cells treated with aspirin under high glucose conditions for 48 h. DCHF-DA staining was used to evaluate intracellular ROS generation. After being treated with high amounts of glucose for 48 h, HK-2 cells were significantly more likely to stain positive for DCHF-DA (green fluorescence). The high glucose-induced increase in DCHF-DA staining was partially weakened by aspirin treatment (Fig 4A). High expression of COX2 protein and mRNA was observed in cells cultured under 30 mmol/L glucose for 48 h. To explain the relevance of COX2 in ferroptosis in vitro, aspirin was used in the experiment. COX2 gene disruption was confirmed by Western blotting analysis and qRT–PCR. After 48 h, COX2 expression considerably increased in high glucose-cultured HK-2 cells, which was attenuated by aspirin administration (Fig 4B). In cells cultured in high-glucose medium with aspirin, the protein expression of SLC7A11, GPX4, and FTH-1 was clearly upregulated, while TFR-1 was decreased compared to cells cultured in high-glucose medium (Fig 4B). These genes exhibited parallel mRNA expression (Fig 4C–4G). In high glucose-cultured HK-2 cells, both Fer-1 treatment and aspirin restored GPX4,

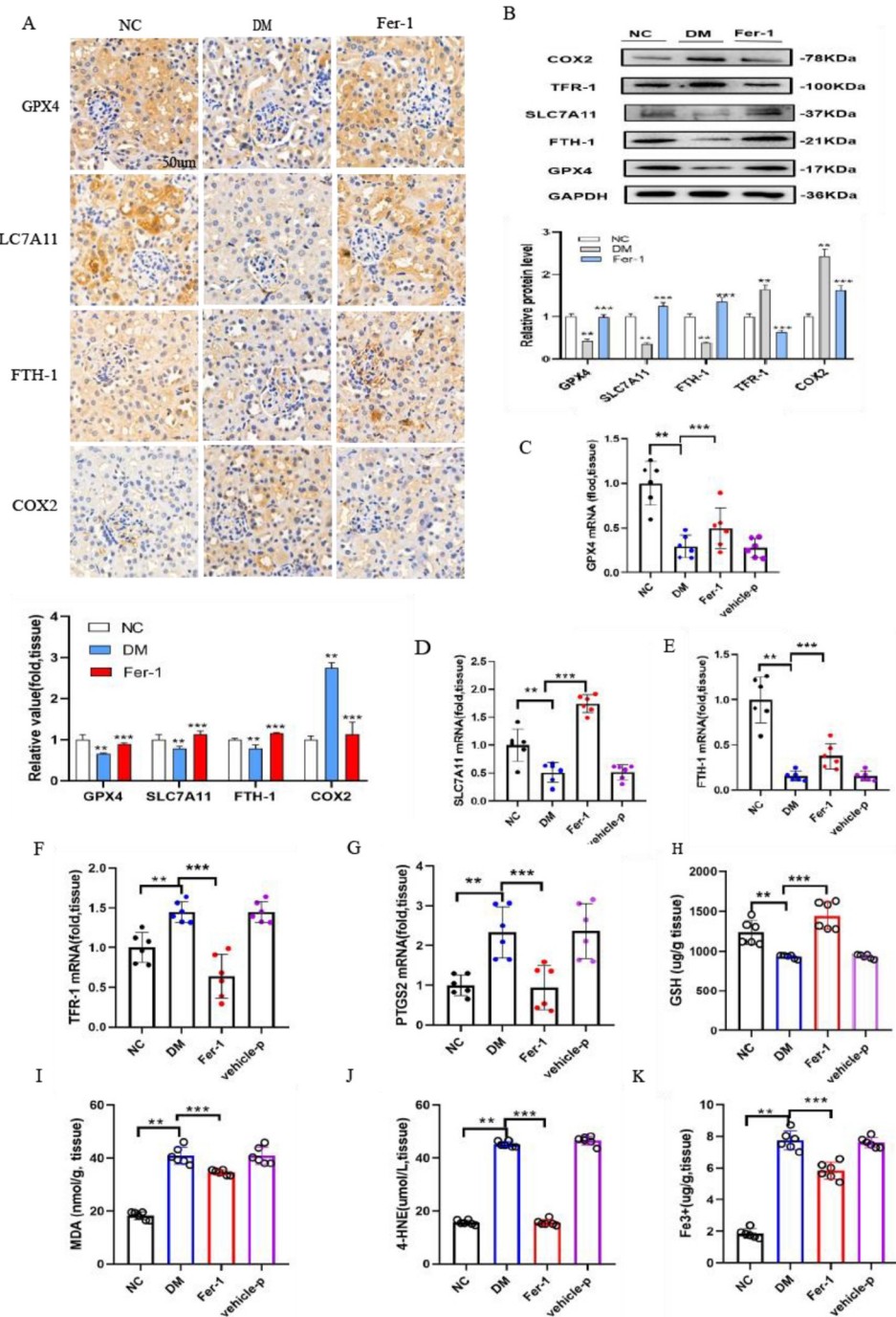

**Fig 3. Ferroptosis-related changes in the kidney are observed in diabetic mice.** (A) Immunohistochemical staining results and semiquantification analyses of GPX4, SLC7A11, FTH-1 and COX2 in the kidneys of nondiabetic, diabetic control and diabetic treated with Fer-1 mice. (B) Western blot analysis and quantification of GPX4, SLC7A11, FTH-1, TFR-1 and COX2 in the kidneys of nondiabetic, diabetic and diabetic mice treated with Fer-1. (C-G) The mRNA expression levels of GPX4 (C), SLC7A11 (D), FTH-1 (E), TFR-1 (F) and PTGS2 (G) in kidney tissue in mice. (H-K) GSH (H), MDA (I), 4-HNE (J) and iron concentrations (K) were detected in the kidney tissue of mice. Data are the mean ± SEM. $^{**}p < 0.05$ *vs.* NC group; $^{***}p < 0.05$ *vs.* DM group.

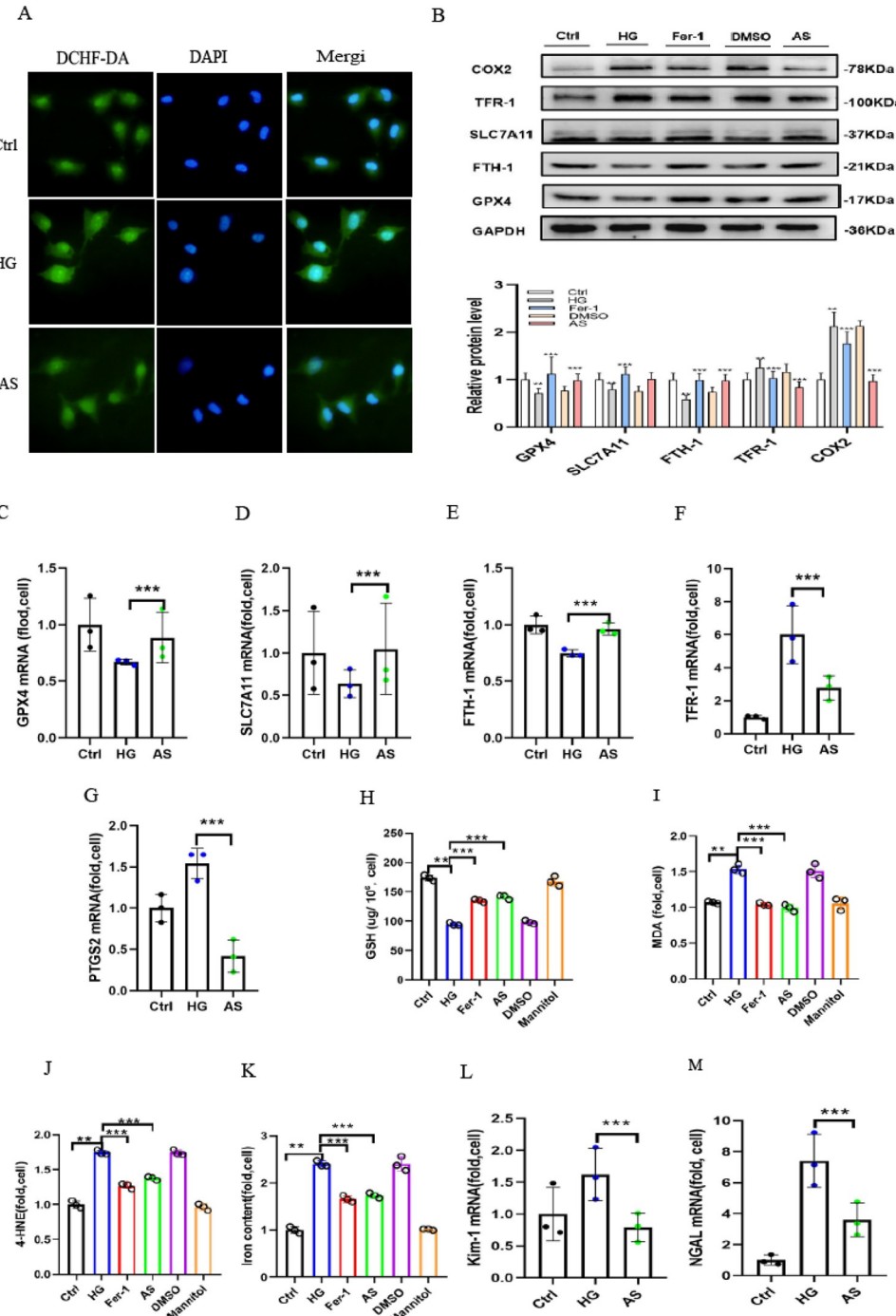

**Fig 4. High glucose-induced ferroptosis was alleviated by treatment with aspirin.** (A) DCHF-DA staining revealed ROS (green) generation in high glucose-cultured HK-2 cells compared to control cells, which was partially alleviated by AS treatment. (B) Western blot analysis of GPX4, SLC7A11, FTH-1, TFR-1 and COX2 protein expression in cells in the Ctrl, HG, Fer-1, DMSO and AS groups. (C-G) GPX4 (C), SLC7A11 (D), FTH-1 (E), TFR-1 (F) and PTGS2 (G) mRNA expression in cells in the Ctrl, HG and AS groups. (H-K) GSH (H), MDA (I),4-HNE(J) and iron concentrations (K) of cells in the Ctrl, HG and AS groups. (L-M) The KIM-1 (L) and NGAL (M) increased mRNA levels induced by high glucose were relieved by treatment with AS. Data are the mean ± SD. **$p < 0.05$ *vs.* Ctrl group; ***$p < 0.05$ *vs.* HG group.

SLC7A11, FTH-1 and TFR-1 expression. These results demonstrate that COX2 hyperactivation is associated with ferroptosis in high glucose-cultured HK-2 cells. Normally, untreated control cells contained only a minimal amount of MDA and 4-HNE. The cells treated with high glucose showed the maximal level compared to NC group cells. Interestingly, aspirin treatment markedly attenuated the level compared to that in HG group cells (Fig 4I and 4J). The obtained data showed that GSH was also clearly decreased and that the levels of iron were higher in HG group cells. However, the changes were ameliorated by Fer-1 and aspirin in high-glucose medium (Fig 4H and 4K). The abovementioned data showed that the inhibition of COX2 ameliorated the ferroptosis sensitivity of HK-2 cells under high glucose conditions. Moreover, the mRNA levels of KIM-1 and NGAL were significantly increased in high glucose-cultured HK-2 cells, and these changes were alleviated by Fer-1 and aspirin (Fig 4L and 4M).

## 3.5 Transfection of COX2-siRNA inhibits ferroptosis in high glucose-cultured HK-2 cells

Ferroptosis is a newly discovered iron-dependent form of nonapoptotic cell death; when GSH is depleted or GPX4 is inhibited, ferroptosis can be accelerated. To explore the underlying mechanism of ferroptosis induced by COX2, we examined the genes associated with ferroptosis. Immunofluorescence microscopy showed intense fluorescence correlated with GPX4 expression in control HK-2 cells. Of note, decreased fluorescence was detected in high glucose-cultured HK-2 cells compared to the controls. Interestingly, GPX4-related levels were partially restored following the transfection of COX2-siRNA in high glucose-cultured cells (Fig 5A). Furthermore, we examined the GSH concentration and GPX4 activity in HK-2 cells. Western blotting analysis showed that GPX4 expression was decreased in high glucose-cultured HK-2 cells (Fig 5B). Similarly, FTH1 was significantly decreased in high glucose-cultured HK-2 cells. This contention led to the investigation of FTH1 and GPX4 degradation in ferroptosis in HK-2 cells cultured in high glucose. However, COX2-siRNA treatment reversed the expression to levels similar to those in control cells (Fig 5B). The mRNA expression of these genes displayed parallel results, and they revealed that transfection of COX2-siRNA significantly affected FTH1 and GPX4 degradation in HK-2 cells after high glucose treatment (Fig 5C–5G). To explore whether COX2 is a target of ferroptosis, we detected the levels of GSH. The obtained results indicated that the GSH levels were decreased after high glucose treatment, which was partially restored by COX2-siRNA transfection. Understandably, this study has demonstrated that COX2 gene disruption can block such depletion of GSH (Fig 5H). Thus, higher levels of MDA and 4-HNE were also observed in high glucose-treated HK-2 cells, and this decrease was ameliorated in high glucose-cultured cells transfected with COX2-siRNA (Fig 5I and 5J). Ferroptosis and lipid hydroperoxidation require free iron, and the obtained data further showed that the intracellular free iron concentration was significantly increased in high glucose-cultured HK-2 cells, which was attenuated by COX2 gene disruption for 48 h (Fig 5K). Meanwhile, the mRNA levels of KIM-1 and NGAL increased in high glucose-cultured HK-2 cells. Furthermore, these changes were also attenuated by COX2-siRNA transfection (Fig 5L and 5M).

## 3.6 RSL-3-induced ferroptosis was ameliorated by Fer-1 and aspirin

It has been reported that RSL3 is a GPX4 inhibitor. We utilized RSL3 to cause ferroptosis in HK-2 cells in order to further explore the role of COX2 in ferroptosis. Compared with the Ctrl group, RSL-3-triggered ROS accumulation exhibited stronger fluorescence intensity, while the high degree of fluorescence was strongly alleviated by aspirin (Fig 6A). Furthermore, the viability of cells treated with RSL-3 was assessed by CCK8 assay. Our study showed that the RSL-

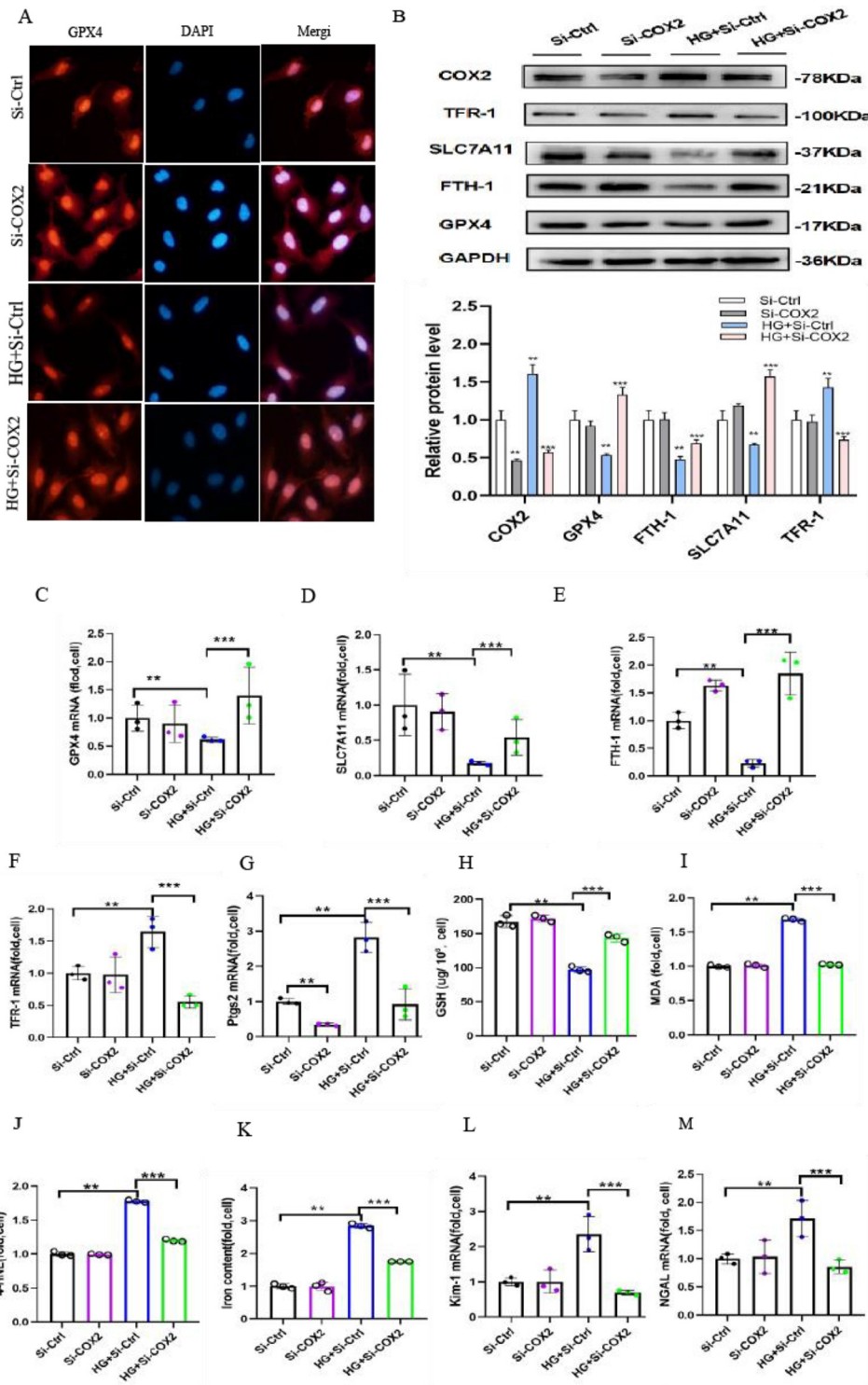

**Fig 5. Knockdown of COX2 ameliorated the ferroptosis sensitivity of cells cultured in high glucose.** (A) Immunofluorescence analysis with GPX4 antibody (red) and nuclear staining with DAPI (blue) in the HK-2 cells of the Si-Ctrl group, Si-COX2 group, HG+Si-Ctrl group and HG+Si-COX2 group. (B) Western blotting showed the protein expression of COX2, GPX4, SLC7A11, FTH-1 and TFR-1 in the cells of each group. G) GPX4 (C), SLC7A11 (D), FTH-1 (E), TFR-1 (F) and PTGS2 (G) mRNA expression in cells in each group. (H-J) GSH (H), MDA (I), 4-HNE (J)and iron concentrations (K) of cells in each group. (K-L) KIM-1 (L) and NGAL (M) increased mRNA expression was alleviated by transfection of COX2-siRNA in high glucose. Data are expressed as the mean ± SD. $^{**}p < 0.05$ *vs.* Si-Ctrl group; $^{***}p < 0.05$ *vs.* HG+Si-Ctrl group.

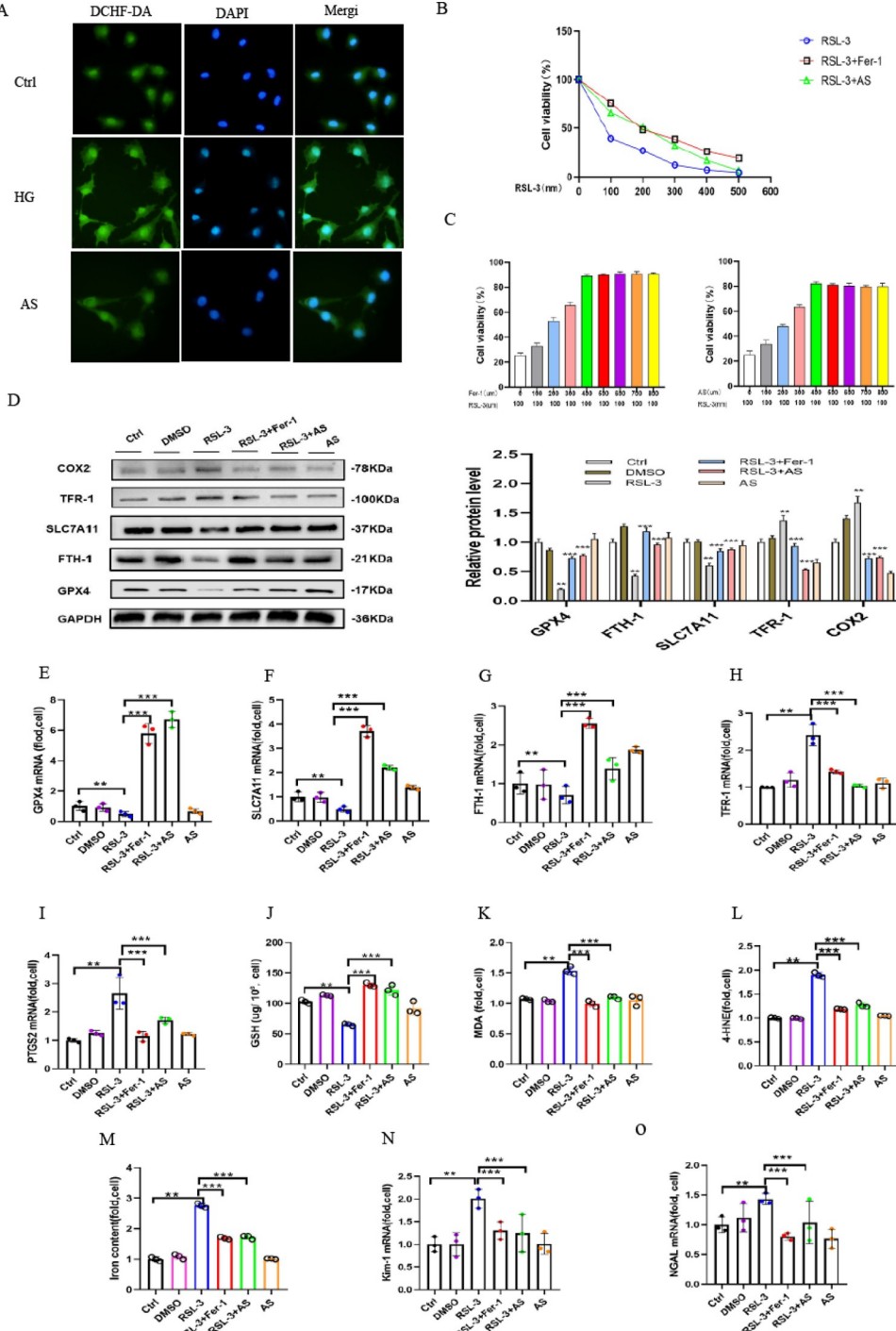

**Fig 6. Cellular ferroptosis induced by RSL-3 was alleviated by Fer-1 and aspirin.** (A)DCHF-DA staining revealed the generation of ROS (green) in the HK-2 cells cultured in 100 nmol/L RSL-3 medium compared to the control cells, and it was partially reduced by aspirin. (B) Cell viability was measured in HK-2 cells cultured with different concentrations of RSL-3 for 48 h, and it was partially improved by 400 nm Fer-1 or 400 μm aspirin treatment. (C) Cell viability measurements in HK-2 cells treated with RSL-3 100 nm for 48 h were partially improved by treatment with different concentrations of Fer-1 or AS. (D) Western blot analysis of SLC7A11, GPX4, FTH-1, TFR-1 and COX2 protein expression of cells in Ctrl, DMSO, RSL-3 100 nm, RSL-3 100 nm+Fer-1 400 nm, RSL-3 100 nm+AS400 μm and AS 400 μm groups. (E-I) GPX4 (E), SLC7A11 (F), FTH-1 (G), TFR-1 (H) and PTGS2 (I) mRNA expression in cells of each group. (J-M) GSH (J), MDA (K), 4-HNE(L)and iron concentrations (M) in the cells of each group. (N-O) The increased mRNA expression of KIM-1 (N) and NGAL (O) induced by RSL-3 was relieved by treatment with Fer-1 or aspirin. Data are expressed as the mean ± SD. $^{**}p < 0.05$ *vs*. Ctrl group; $^{***}p < 0.05$ *vs*. RSL-3 group.

3 concentration increased as fluorescence intensity was clearly decreased. When the RLS-3 concentration was over 0.1 μM, it clearly decreased. As HK-2 cells were treated with different concentrations of RSL-3 for 48 h, cell viability was partially improved by 400 nm Fer-1 or 400 μm aspirin treatment (Fig 6B). The cell viability measurements in HK-2 cells treated with RSL-3 100 nm for 48 h were partially improved by treatment with different concentrations of Fer-1 or aspirin (Fig 6C). Meanwhile, treatment with Fer-1 or aspirin inhibited the COX2-increased expression of RSL-3-treated cells (Fig 6D). Decreased mRNA expression of SLC7A11, GPX4 and FTH-1 and increased expression of PTGS2 and TFR-1 caused by ferroptosis were clearly reversed by Fer-1 or aspirin (Fig 6E–6I). In addition, our study showed that GSH was also clearly decreased in RSL-3-treated cells (Fig 6J). Moreover, lipid peroxidation product accumulation, MDA (Fig 6K), 4-HNE(Fig 6L)and concentrations of iron (Fig 6M) under ferroptosis conditions were also attenuated after Fer-1 or aspirin treatment. Otherwise, the mRNA levels of KIM-1 and NGAL increased in RSL-3-treated HK-2 cells. Interestingly, these changes were also mitigated by Fer-1 or aspirin (Fig 6N and 6O). Overall, these data demonstrated that aspirin could alleviate the pathogenesis of ferroptosis by downregulating COX2 expression.

## 3.7 Transfection of COX2-siRNA alleviates RSL-3-induced ferroptosis in HK-2 cells

To further verify that transfection of COX2-siRNA may protect against ferroptosis-induced RSL-3 in HK-2 cells, intense GPX4 fluorescence in HK-2 cells was revealed by immunofluorescence microscopy. Notably, decreased fluorescence was detected in RSL-3-treated HK-2 cells compared to the controls. In contrast, GPX4-related levels were partially recovered by the transfection of COX2-siRNA in RSL-3-treated HK-2 cells (Fig 7A). Furthermore, CCK8 assays showed that RSL-3 induced massive cell death through the pathogenesis of ferroptosis after 48 h. Interestingly, RSL-3-induced ferroptosis-specific cell death was attenuated by COX2 gene disruption. The CCK8 assay demonstrated that HK-2 cell death induced by RSL-3 was alleviated by COX2-siRNA treatment (Fig 7B). Thus, the high expression of COX2 protein (Fig 7C) and mRNA (Fig 7H) was also detected in cells cultured with 100 nm RSL3 for 48 h. We also detected that ferroptosis-related protein expression was reversed by transfection of COX2-siRNA. Our study showed that SLC7A11, GPX4, and FTH-1 protein expression was clearly increased and TFR-1 was decreased in cells transfected with COX2-siRNA (Fig 7C). The mRNA expression of these genes showed parallel results (Fig 7D–7G). The obtained data revealed that GSH was also clearly decreased in RSL-3-treated cells. However, the changes were ameliorated by COX2 knockdown in RSL-3-treated cells (Fig 7I). Simultaneously, pretreatment with transfection of COX2-siRNA reduced the concentrations of MDA (Fig 7J), 4-HNE (Fig 7K)and iron (Fig 7L), which implied that transfection of COX2-siRNA could improve the states of lipid peroxidation and iron overload induced by cells cultured in 100 nm RSL-3 medium. In addition, the mRNA levels of KIM-1 and NGAL increased in RSL-3-treated HK-2 cells. Interestingly, this pathogenesis was also impaired by transfection with COX2-siRNA (Fig 7M and 7N). Overall, the abovementioned research indicated that the inhibition of COX2 can ameliorate the ferroptosis sensitivity of RSL-3-treated HK-2 cells.

## 3.8 COX2 gene inhibition prevents renal tubular injury in diabetic kidney disease

The diabetic mice received intragastric treatment with aspirin and 0.5% Na-CMC. HE staining revealed that aspirin ameliorated the renal morphological changes in diabetic mice, including tubular epithelial degeneration with brush border loss and dilatation of the tubular lumen.

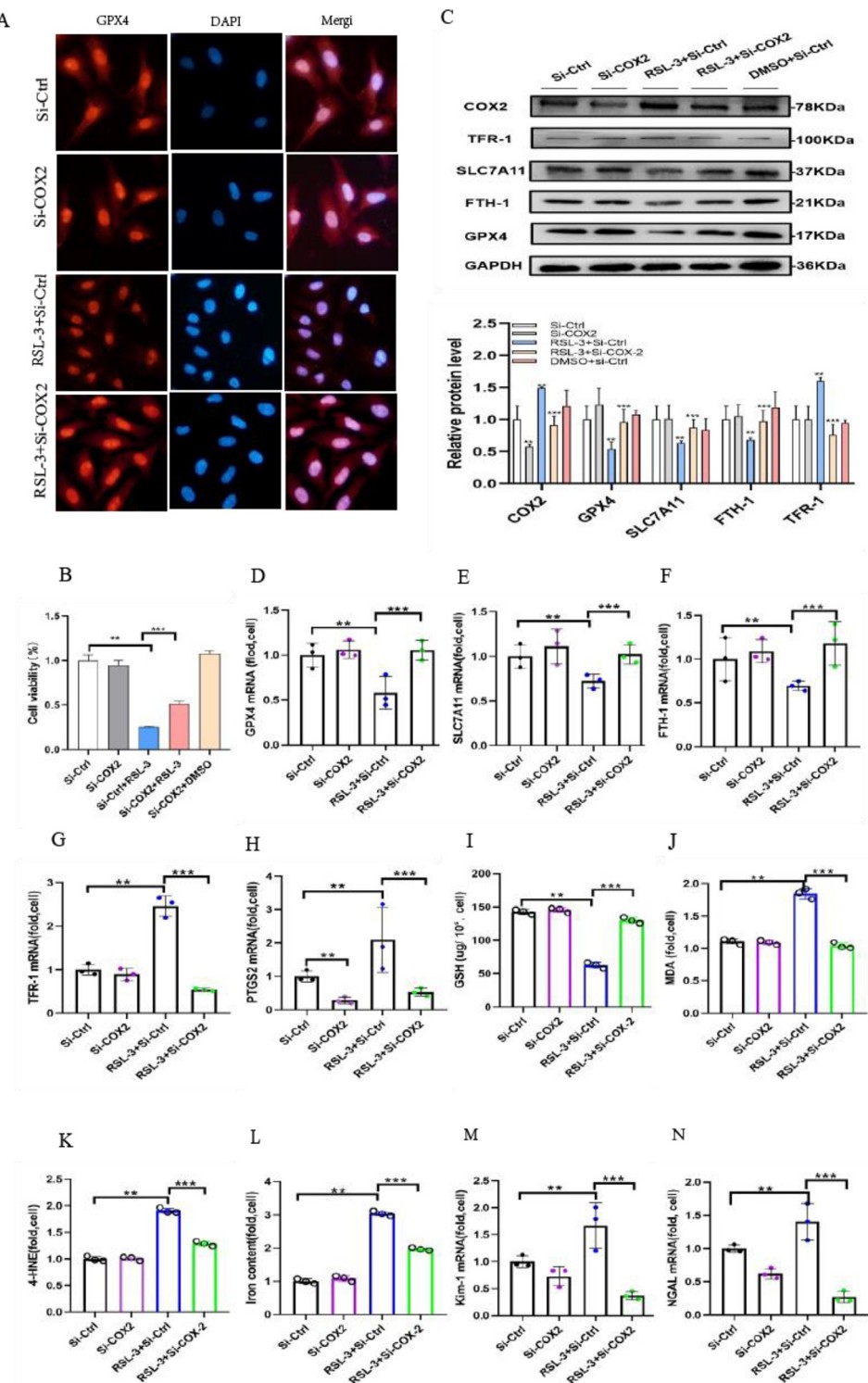

**Fig 7. Cellular ferroptosis-induced RSL-3 was inhibited by transfection with COX2-siRNA.** (A)
Immunofluorescence analysis with GPX4 antibody (red) and nuclear staining with DAPI (blue) of HK-2 cells in the Si-
Ctrl, Si-COX2, RSL-3+Si-Ctrl, RSL-3+Si-COX2 and DMSO+Si-Ctrl groups. (B) The cell viability measurements in
HK-2 cells treated with RSL-3 100 nm for 48 h were partially improved by transfection of COX2-siRNA. (C) Western
blotting results of COX2, GPX4, SLC7A11, FTH-1 and TFR-1 in cells in each group. (D-H) The mRNA expression of
GPX4 (D), SLC7A11 (E), FTH-1 (F), TFR-1 (G) and PTGS2 (H) in each group of cells. (I-L) GSH (I), MDA (J), 4-HNE
(K) and iron concentrations (L) of HK-2 cells in the Si-Ctrl, Si-COX2, RSL-3+Si-Ctrl and RSL-3+Si-COX2 groups.

(M-N) KIM-1 (M) and NGAL (N) increased mRNA expression induced by RSL-3 and were relieved by transfection of COX2-siRNA. Data are the mean ± SEM. **$p < 0.05$ *vs.* Si-Ctrl group; ***$p < 0.05$ *vs.* Si-Ctrl+ RSL-3 group.

PAS staining demonstrated mitigated tubular epithelial disruption, and Masson staining revealed reduced relative collagen deposition and thickening of the basement membrane in aspirin-treated diabetic mice (Fig 8A). This study showed that the levels of blood glucose (Fig 8B) and body weight (Fig 8C) were not different in the DM, AS and vehicle-G groups. Aspirin had no effect on the blood glucose and body weights of diabetic mice, which indicated that aspirin did not affect the regulation of blood glucose in diabetes. Polyuria was clearly ameliorated by aspirin in diabetic mice. The average 24-h urine volume was approximately 14 mL in the DM group but approximately 6 mL in the AS group (Fig 8D). Compared to diabetic mice, urinary albuminuria in the AS group (Fig 8E) was almost 2 times lower than that in diabetic mice, the ACR in the AS group (Fig 8F) decreased by almost 50%, and the kidney weight (Fig 8G) decreased from 9.93 to 7.9 mg/g. Even though these exponents of renal injury were still higher in the AS group than in the normal group, they were clearly ameliorated compared with those in diabetic mice. Moreover, compared to the diabetic mice, decreased NGAL and KIM-1 mRNA levels were detected in aspirin-treated diabetic mice, demonstrating alleviation of renal pathological damages in diabetic mice (Fig 8H and 8I).

### 3.9 COX2 gene inhibition attenuates ferroptosis in diabetic kidney disease

Immunohistochemical staining showed that GPX4 and COX2 were primarily expressed in renal tubular epithelium. Notably, changes in COX2 expression were opposite compared to those detected for GPX4 in diabetic mice. The obtained data showed that COX2 expression was markedly increased in diabetic mice (Fig 9A). Both the protein and mRNA levels of GPX4 and COX2 were explored to evaluate ferroptosis in vivo. Western blotting analysis and RT–PCR studies showed that GPX4 expression was decreased in diabetic mice. GPX4 expression was significantly increased in aspirin-treated diabetic mice. In contrast to GPX4, COX2 expression in diabetic mice was increased, while a decrease was noted in aspirin-treated diabetic mice (Fig 9B). Subsequently, the transcription conditions of FTH1 and TFR-1 in the kidneys were assessed by RT–PCR analyses. In diabetic kidneys, decreased FTH1 and increased TFR-1 mRNA levels were detected, suggesting that there may be a high intracellular free iron concentration (Fig 9E, 9F and 9K). Interestingly, in AS group mice, GPX4, SLC7A11 and FTH-1 protein expression were significantly enhanced, and COX2 expression was clearly decreased in the renal sections compared with the DM group (Fig 9B). The mRNA expression of the genes was similar to the corresponding immunohistochemical staining (Fig 9C–9G). The GSH levels were clearly lower in diabetic mice than in controls, whereas the GSH levels were increased in AS mice, indicating that the ability of the kidneys to resist oxidative stress was ameliorated in AS group mice (Fig 9H). To assess lipid hydroperoxidation during the process of diabetic-induced injury, the extent of 4-HNE and MDA were observed. The levels of MDA and 4-HNE were notably increased in the kidneys of diabetic mice. Both of them decreased by approximately 50% in aspirin-treated diabetic mice compared to diabetic mice (Fig 9I and 9J). Moreover, we observed that the iron accumulation decreased by almost 65% in the kidneys of AS group mice and was ameliorated (Fig 9K). Ultrastructural analysis showed that the changes in mitochondrial morphology in HK-2 cells exposed to high glucose, including increased membrane density with shrunken mitochondria and mitochondrial ridge decrease or even disappearance, were alleviated by aspirin. The mitochondrial ridge remained visible, and the status of increased membrane density and mitochondrial shrinkage was weakened in the AS group (Fig 9L). Overall, this study revealed that the mitochondrial

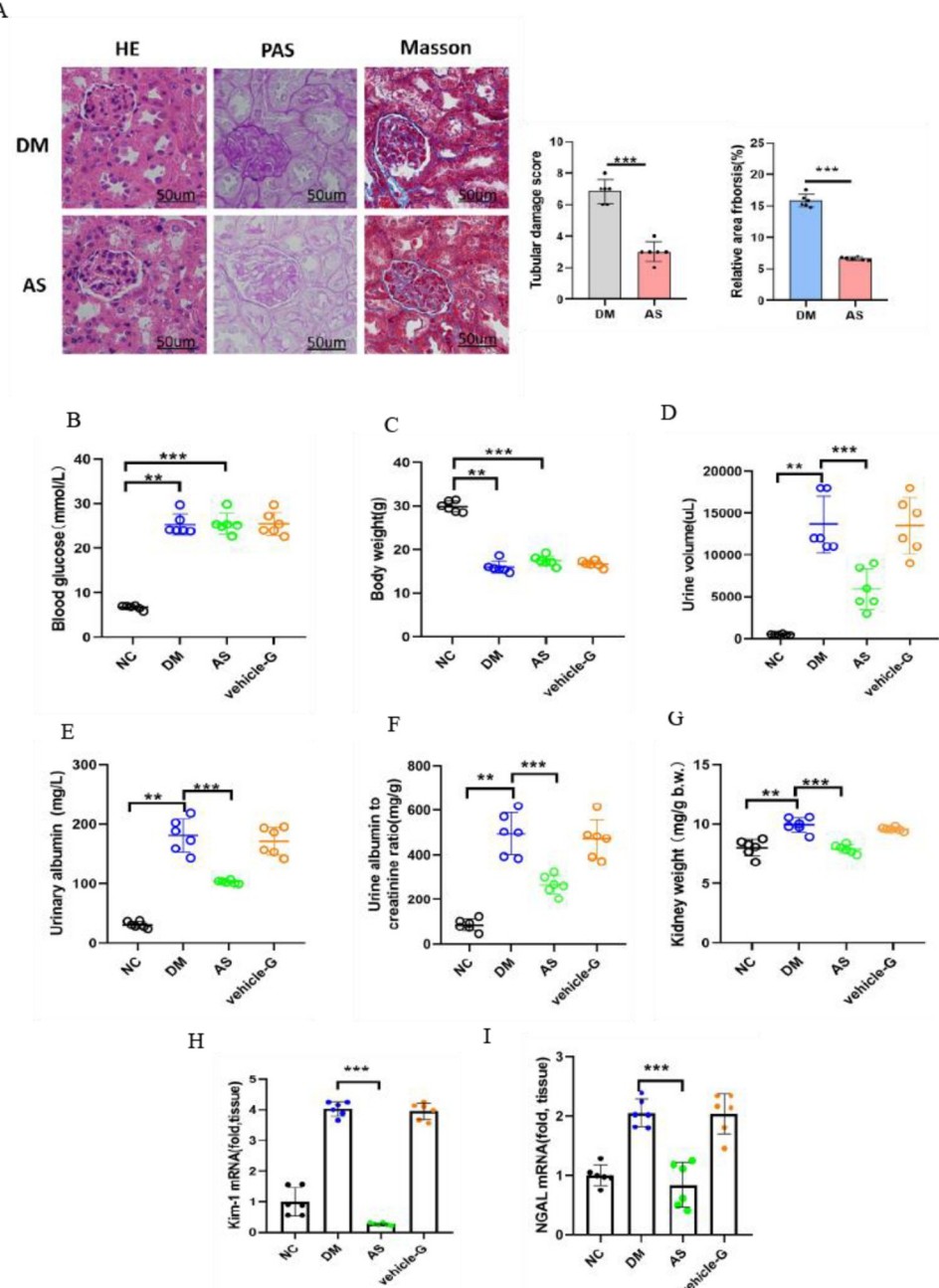

**Fig 8. Aspirin ameliorates renal injury in diabetic mice.** (A) HE staining, PAS staining and Masson staining of kidneys in nondiabetic and diabetic control AS mice were assessed. Relative collagen (%) and relative area of fibrosis (%) were measured by the Image J program. (B-G) Physiological parameters, including blood glucose (B), body weight (C), 24-h urine volume (D), urinary albuminuria (E), albumin-to-creatinine ratio ACR (F) and kidney weight/body weight (G) were measured. (H-I) KIM-1 (H) and NGAL (I) increased mRNA expression induced by diabetes and were relieved by aspirin treatment. Data are the mean ± SEM. **$p < 0.05$ *vs*. NC group; ***$p < 0.05$ *vs*. DM group.

morphological pathogenesis caused by diabetic ferroptosis can be ameliorated by aspirin. The biomarkers of diabetic ferroptosis were observed in HK-2 cells cultured with high glucose, which was significantly improved by aspirin. The abovementioned data demonstrated that aspirin significantly improved renal injuries in diabetic mice. Therefore, our research

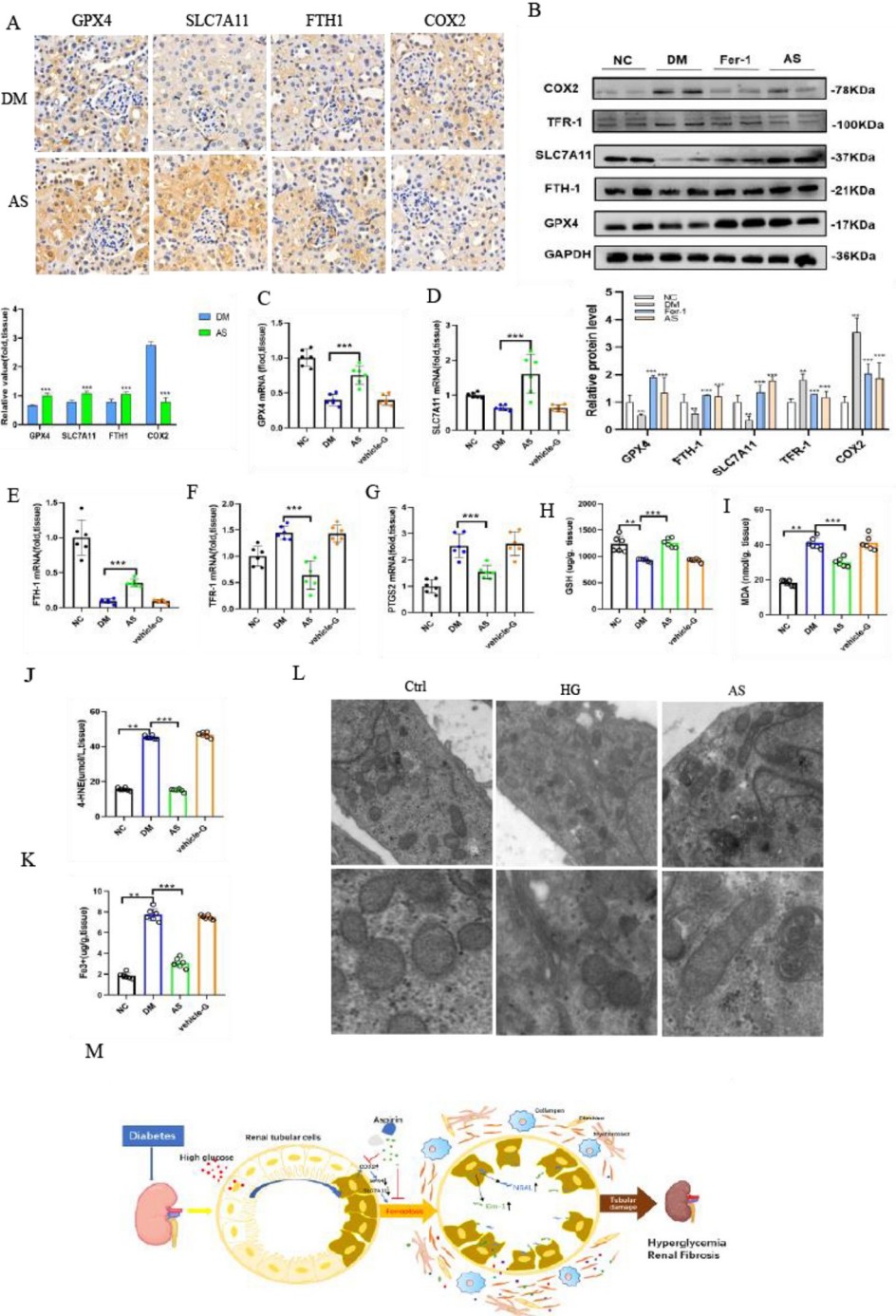

**Fig 9. Aspirin ameliorates ferroptosis indices in the diabeitc kidney.** (A) GPX4, SLC7A11, FTH-1 and COX2 of the kidneys in nondiabetic, diabetic control and diabetic treated with AS mice were assessed based on immunohistochemical staining results and semiquantification analyses. (B) Western blot analysis of GPX4, SLC7A11, FTH-1, TFR-1 and COX2 in the kidneys of nondiabetic, diabetic control and diabetic mice treated with AS. (C-G) GPX4 (C), SLC7A11 (D), FTH-1 (E), TFR-1 (F) and PTGS2 (G) mRNA expression in the kidney tissue of mice were measured. (H-K) The GSH (G), MDA (H), 4-HNE (I) and iron concentrations (J) in the kidney tissue were analyzed. (L) The mitochondrial morphology of cells was detected by transmission electron microscopy in the Ctrl, HG and AS groups. (M) Schematic model: COX2/GPX4-mediated ferroptosis in the process of diabetic kidney. Data are the mean ± SEM. $^{**}p < 0.05$ *vs*. NC group; $^{***}p < 0.05$ *vs*. DM group.

illustrated that the pharmacological mechanism of aspirin ameliorates ferroptosis of the kidney in diabetic mice via inhibition of COX2, as shown in Fig 9M.

## 4. Discussion

DKD is a potentially life-threatening microvascular complication of diabetes mellitus (DM) [32]. DKD is closely related to several types of cell death, including apoptosis, necroptosis and ferroptosis. These studies showed that tubular cell death may be linked to tubular pathological changes and renal fibrosis in prolonged DKD. The majority of them have been thoroughly investigated except for ferroptosis, which is a more recent discovery [33,34]. The term ferroptosis is typically characterized by reduced antioxidant defense, iron overload and lipid peroxidation [35–37]. The essential function of cell metabolism, especially phospholipid peroxidation, in ferroptosis depends on iron. It has been reported that system xc-, GSH synthesis, and glutathione peroxidase 4 (GPX4) can protect cells against death induced by oxidative stress conditions, especially that caused by inhibition of system xc- activity [38–42]. GPX4 is a selenoprotein of antioxidant enzymes and the major enzyme capable of catalyzing the reduction of PLOOHs in mammals. Furthermore, GPX4 is essential for the survival of proximal kidney tubular cells, and the inducible depletion of GPX4 causes massive ferroptosis of renal tubular epithelial cells. Ferroptosis is essential in the genetic model of inducible whole-body depletion of Gpx4 and in folic acid-induced acute kidney injury (AKI) [18,19].

Although the role of ferroptosis in tubular injury in DKD has been less researched, it has been demonstrated that iron homeostasis is disrupted in the kidneys of diabetic mice [43]. Through the Fenton reaction, iron can directly produce excessive ROS, thus causing further increasing oxidative damage. Because iron is a critical component of cell viability and death, cellular iron homeostasis must be carefully managed. Our studies showed that TFR-1 expression was clearly increased, and FTH-1 expression was significantly decreased at both the protein and mRNA levels, indicating that iron homeostasis is imbalanced in diabetes. The evident release of free iron, which is a typical characteristic of ferroptosis, has been observed in high glucose-cultured HK-2 cells. We demonstrated that diabetic nephropathy was significantly attenuated by Fer-1. This study also showed that the increased mRNA levels of KIM-1 and NGAL in high glucose-cultured HK-2 cells were attenuated by Fer-1, indicating that ferroptosis plays an important role in cellular injury. Overall, these data demonstrated that ferroptosis inhibition is operative in DKD.

Even though tubular injury plays an important role in DKD pathogenesis, there is currently no cell-specific therapy for the condition. Evidence continues to accumulate showing the link between kidney disease and disturbances of tubular cell ferroptosis, and such insight will open up potential for developing new therapeutic approaches for DKD patients. Our study demonstrated the biological functions of COX2 in tubular cells. Importantly, COX2 in the kidney was markedly upregulated in DKD, indicating that COX2 is a potential target for DKD. Our results showed that tubular cells secrete COX2 under diabetic conditions. This research was designed to explore the biological functions of COX2 in tubular cells. We found that downregulating COX2 significantly lessened urinary albumin excretion and tubular damage in aspirin-treated diabetic mice, demonstrating that COX2 may be an essential target molecule for tubular damage.

COX2 is a proximal tubular-specific enzyme. Similarly, this study revealed that ferroptosis occurred in damaged PTECs. Our work showed that downregulation of COX2 alleviates tubular redox injury in DKD, but its role in ferroptosis remains enigmatic. ROS accumulation induced by hyperglycemia supplies oxidative stress. GPX4 and GSH were reduced in diabetic kidneys, which further exacerbated the oxidative stress response. As COX2 expression

increased, it was examined in the kidneys of diabetic mice, which was in agreement with previously reported studies [20,25,26]. Functionally, our study verified one unappreciated role of COX2 in mediating renal tubular cell ferroptosis. In this study, we demonstrated whether diabetic ferroptosis could be suppressed by the downregulation of COX2 expression, thereby showing that COX2 is a potential target for the progression of DKD. First, our study demonstrated that the sensitivity of HK-2 cells to ferroptosis was decreased by the specific knockdown of COX2 under high glucose conditions. When COX2 was suppressed, TFR-1 expression was decreased, while its target gene FTH-1 expression was increased. As a result of silencing COX2, cells grown in high glucose medium have an enhanced antioxidant capacity, as well as GPX4 and SLC7A11 expression. Knockdown of COX2 diminished free iron accumulation in cells, increased resistance to oxidative stress and weakened lipid peroxidation occurrence, which induced ferroptosis and further decreased cell injury under high glucose conditions. In addition, as a biomarker of ferroptosis sensitivity, the intracellular levels of GSH were lower in high glucose-cultured HK-2 cells and diabetic mice even without aspirin treatment, suggesting that downregulation of COX2 expression can inhibit the occurrence of diabetic ferroptosis.

It is difficult to investigate the process of ferroptosis due to the lack of specific markers. Accordingly, ferroptosis is characterized by a pronounced relentless process of lipid hydroperoxidation and iron overload [37]. Lipid hydroperoxides are the executors of ferroptosis, which causes massive cell death [44], while increased intracellular free iron can catalyze ferroptosis [45]. MDA and 4-HNE are the end products of lipid hydroperoxides [14]. They were used to evaluate the extent of lipid hydroperoxidation in this research. Under high glucose conditions in vivo and in vitro, our studies showed that the increase in MDA and 4-HNE levels were dampened by COX2 gene disruption. In addition, the Iron Assay Kit assay revealed increased intracellular free iron levels, which were attenuated by COX2 gene disruption in HK-2 cells cultured under high glucose conditions. Ferroptosis is important in HK-2 cells cultured with high glucose. We hypothesized that COX2 can modulate the pathogenesis of ferroptosis. This study revealed that downregulation of COX2 was beneficial to cellular homeostasis during damage to HK-2 cells and DKD. In this study, RSL-3 directly suppressed GPX4 signaling in cultured HK2 cells, and we demonstrated that the COX2 expression profile can modulate significant ferroptosis-specific cell death induced by RSL-3.

Although the role of aspirin-mediated organ protection has been greatly emphasized, the exact role of aspirin has long been controversial. Diabetic mice developed albuminuria and tubular injury, showing that this model is likely to be an ideal experimental model to study tubular injury-related kidney damage induced by hyperglycemia. Aspirin can ameliorate albuminuria in this model. In our study, aspirin treatment under diabetic conditions decreased COX2 levels and showed renoprotection. Thus, elevated GSH levels may be an aspirin-mediated renoprotective effect in diabetic mice. Systemic improvements, such as GSH, are likely to have a protective effect against tubular loss and renal fibrosis.

Ordinarily, the key antioxidases GPX4 and GSH catalyze noxious lipid hydroperoxides into harmless lipid alcohols [12]. Thus, GPX4 inhibitors and system Xc- antagonists (related to GSH depletion) can act as inducers of ferroptosis [11]. Subsequently, under high glucose conditions, we demonstrated that COX2 expression could be downregulated by aspirin, which resulted in SLC7A11, GPX4, FTH-1 and TFR-1 expression being also altered, and COX2 knockdown ameliorated the process of ferroptosis in cells. To confirm the effect of aspirin on ferroptosis, we applied it to cells induced by RSL-3 and detected that COX2 expression decreased and the biomarkers of ferroptosis improved. When aspirin was used to treat diabetic mice, we observed that the increase in COX2 was inhibited in mice because the pathogenesis of ferroptosis was also ameliorated. In diabetic mice, aspirin also mitigated pathological kidney

damage. Thus, our study demonstrated that aspirin can upregulate SLC7A11 and GPX4 expression by suppressing COX2. Moreover, the antioxidative stress ability was improved in the kidneys of diabetic mice. Consequently, the iron deposition in the kidneys of diabetic mice was reduced by increasing FTH-1 and downregulating TFR-1. Because iron metabolism stabilized and antioxidant capacity was improved in diabetic mice, ferroptosis of the kidneys was mitigated. Herein, a significant decrease in "GPX4 activity" was observed in diabetic mice, while it was enhanced in diabetic mice treated with aspirin. The abovementioned findings showed that the depletion of GSH and oxidative stress caused by high glucose were ameliorated by downregulating COX2. Here, our study shows that aspirin can effectively inhibit COX-2-mediated ferroptosis in vitro and reduce subsequent renal injury responses in vivo. Accordingly, inhibition of ferroptosis caused by downregulation of COX2 could alleviate the progression of DKD. An explanation for the nephroprotective effects of aspirin was provided by our study, i.e., inhibition of ferroptosis by downregulating COX2.

Therefore, this study demonstrated a novel link between COX2 inhibition, ferroptosis, and DKD progression. While there are multiple mechanisms by which aspirin exerts renoprotective effects, this study indicated that cellular ferroptosis plays a major role in the progression of DKD, while alleviating ferroptosis by inhibiting the COX2 enzyme can potentially attenuate the progression and development of DKD. As a result of these novel insights into DKD pathogenesis, therapeutic strategies that alleviate renal tubular injury can be developed and provide a new pharmacological target for preventing and treating DKD.

## Supporting information

**S1 Table.**
(DOCX)

**S1 Raw images.**
(PDF)

## Author Contributions

**Data curation:** Dan Li, Dingyuan Tian.

**Funding acquisition:** Zhongming Wu.

**Methodology:** Ziyu Wu.

**Supervision:** Xuejun Liu.

**Writing – original draft:** Ziyu Wu.

**Writing – review & editing:** Zhongming Wu.

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
