## [Decision Letter · Decision Letter 0]

21 Apr 2022

PONE-D-21-40203Aspirin Mediates Protection from Diabetic Kidney Disease by Inducing ferroptosis InhibitionPLOS ONE

Dear Dr. Wu,

Thank you for submitting your manuscript to PLOS ONE. After careful consideration, we feel that it has merit but does not fully meet PLOS ONE’s publication criteria as it currently stands. Therefore, we invite you to submit a revised version of the manuscript that addresses the points raised during the review process.

While both reviewers enjoyed reading the manuscript and found it meaningful, some concerns need to be addressed. These concerns include the detection of mitochondrial specific ROS, the rationale for the amount of aspirin used, poor quality western blot images, and grammatical errors in the manuscript.

We look forward to receiving your revised manuscript.

Kind regards,

Aldrin V. Gomes, Ph.D.

Academic Editor

PLOS ONE

Journal Requirements:

In your cover letter, please note whether your blot/gel image data are in Supporting Information or posted at a public data repository, provide the repository URL if relevant, and provide specific details as to which raw blot/gel images, if any, are not available. Email us at plosone@plos.org if you have any questions

4. In your Methods section, please provide additional information on the animal research and ensure you have included details on : (1) methods of sacrifice (2) methods of anesthesia and/or analgesia, and (2) efforts to alleviate suffering

5. Thank you for submitting the above manuscript to PLOS ONE. During our internal evaluation of the manuscript, we found significant text overlap between your submission and the following previously published works, some of which you are an author.

https://pubmed.ncbi.nlm.nih.gov/31586053/

https://www.nature.com/articles/s41418-021-00755-6

https://linkinghub.elsevier.com/retrieve/pii/S1550413121001704

https://www.nature.com/articles/s41422-020-00441-1?code=c0e202d9-78be-496a-b93a-01b8059a8703&error=cookies_not_supported

https://www.sciencedirect.com/science/article/abs/pii/S0891584920315975?via%3Dihub

https://www.jci.org/articles/view/129903

https://www.cell.com/cell-metabolism/fulltext/S1550-4131(20)30552-0?_returnURL=https%3A%2F%2Flinkinghub.elsevier.com%2Fretrieve%2Fpii%2FS1550413120305520%3Fshowall%3Dtrue

Please revise the manuscript to rephrase the duplicated text, cite your sources, and provide details as to how the current manuscript advances on previous work. Please note that further consideration is dependent on the submission of a manuscript that addresses these concerns about the overlap in text with published work.

"No"

Reviewers' comments:

Reviewer's Responses to Questions

**Comments to the Author**

1. Is the manuscript technically sound, and do the data support the conclusions?

Reviewer #1: Yes

Reviewer #2: Yes

2. Has the statistical analysis been performed appropriately and rigorously? 

Reviewer #1: Yes

Reviewer #2: I Don't Know

3. Have the authors made all data underlying the findings in their manuscript fully available?

Reviewer #1: Yes

Reviewer #2: Yes

4. Is the manuscript presented in an intelligible fashion and written in standard English?

Reviewer #1: No

Reviewer #2: Yes

5. Review Comments to the Author

Reviewer #1: This is a detailed research article that unraveled the role of ferroptosis and its inhibition in diabetic kidney disease(DKD) using in-vitro HK-2 cells and male diabetic mice model. Although interesting, informative, and thorough, a number of concerns remain.

1. In figure 4, mitochondrial ROS generation was supposedly accessed via DCHF-DA staining. How true is this? DCHF-DA ( 2′ ,7′ -dichlorofluorescein diacetate) can be used to detect total intracellular ROS but certainly not mitochondrial specific ROS. I would suggest the authors revise this section and other sections where mitochondrial ROS was mentioned.

2. The authors mentioned using Aspirin, a non-specific inhibitor of COX-2 activation, at 400uM in cells and 50mg/kg in diabetic mice, but there was no rationale on why those concentration were chosen in their experiments. I would suggest they provide some pharmacological reference backing the concentration if its a low, medium, or high concentration.

3. The manuscript focused on about ~5 protein (GPX4, SLC7A11, FTH-1, TFR-1, COX2) expression. Were other antioxidant and oxidative stress protein looked into and what was their results? It would make sense to look into other pathways and include in the supplemental in order to inform readers. (Plus supplemental is missing as I wasn't able to locate it while reviewing).

4. Involvement of mitochondria should be looked into. Since mitochondria are major sources of ROS within the cell and vulnerable to oxidative stress it would make more sense to investigate their involvement in this Fe2+ overload model of DKD. I would suggest the authors include mitochondrial ROS/superoxide, morphological changes, mitochondrial permeability transition, and oxphos protein expression if possible.

5. Western blotting images could be improved especially SLC7A11 protein expression.

6. English expression needs major expression. I would suggest that grammatical errors in the manuscript be corrected.

Reviewer #2: Reviewers' comments:

In the present study, Wang et al. found that ferroptosis in renal tubular cells was involved in the development of diabetic kidney disease, and COX2 is a potential target for ferroptosis. They also demonstrated the beneficial effect of aspirin by inhibiting the diabetes-related ferroptosis. The study is well designed and of promising practical guidance. The manuscript is well written and clearly presented. However, there are some issues that the authors should address and they are as follows:

1. How is the level of 4-HNE in high glucose incubated HK2 cells and together with aspirin? It would be better to have consistent results with the animal model.

2. In Fig 4B, COX2 level is increased with DMSO treatment, is there any explanation for this result?

3. The author showed the high glucose increased the cell death in the manuscript and the beneficial effect of aspirin in the animal model, how is the effect of aspirin to the high glucose induced cell death?

6. PLOS authors have the option to publish the peer review history of their article (what does this mean?). If published, this will include your full peer review and any attached files.

Reviewer #1: No

Reviewer #2: No

---

## [Author Response · Author response to Decision Letter 0]

4 Aug 2022

、I ensure that my manuscript meets PLOS ONE's style requirements, including those for file naming.

2、My blot/gel image data are in Supporting Information.

3、I ensure that I have an ORCID iD and that it is validated in Editorial Manager.

4、All mice were euthanized with 60 mg/ml of pentobarbital sodium at the end of the experimental period.

5、I have revised the manuscript to rephrase the duplicated text.

6、All authors have declared that no competing interests that pertain to this work.This information have be included in my cover letter.

Reviewer #1

Q1:DCHF-DA ( 2′ ,7′ -dichlorofluorescein diacetate) can be used to detect total intracellular ROS, the expression in the text has been modified by us.

Q2:The viability of cells treating with RSL-3 was assessed by CCK8. Our study showed that as RSL-3 concentration increased as it was obviously decreased . When RLS-3 concentration was over 0.1 μM, it decreased obviously. As the HK-2 cells was treated with different concentrations of RSL-3 for 48 h, the cell viability was partially recused by Fer-1 400nm or AS 400um treatment. In Fer-1 group and AS group , cells were respectively cultured in 30 mmol/L glucose medium supplemented with 400nM Fer-1 and 400 μM Aspirin for 48 h .

Q3:In this study, we mainly explored the relationship between aspirin and ferroptosis, so we did not examine other antioxidant and oxidative stress protein.

Q4:Theultrastructural analysis displayed that the changes of mitochondria morphological in the HK-2 cells cultured with high glucose, containing increased membrane density with shrunken mitochondria and mitochondrial ridge decrease or even vanishing, were alleviated by aspirin. The mitochondrial ridge remains obviously visible , the status of increased membrane density and mitochondrial shrinkage were weakened in AS group (Fig 9L). The research implied that the mitochondrial morphological pathogenesis caused by diabetic ferroptosis could be ameliorated by aspirin. 

Q5:I have fixed the grammar mistakes in the article.

Reviewer #2

Q1：In this study, the end products of lipid peroxidation ,MDA,was detected in the cells and animal.

Q2：In Fig 4B, COX2 level is increased with DMSO treatment. Because cells were respectively cultured in 30 mmol/L glucose medium supplemented with 0.1% DMSO for 48 h. 

Q3: In cells being cultured in high-glucose medium with Aspirin,the protein expression of SLC7A11,GPX4, FTH-1 were obviously upregulated while TFR-1 was decreased compared to cells under high-glucose medium ((Fig 4B), these genes exhibited the parallel mRNA expression (Fig 4C-4F).Aspirin treatment markedly attenuated the MDA level of HG group cell(Fig 4I). The GSH was also obviously decreased and the levels of iron was higher in HG group cells,the changes were ameliorated by aspirin in high-glucose medium (Fig 4H,4J).Moreover, the mRNA levels of KIM-1 and NGAL significantly increased in high glucose-cultured HK-2 cells,and these changes were alleviated by aspirin.(Fig 4K-4L)

---

## [Decision Letter · Decision Letter 1]

15 Sep 2022

PONE-D-21-40203R1A spirin Mediates Protection from Diabetic Kidney Disease by Inducing ferroptosis InhibitionPLOS ONE

Dear Dr. Wu,

Thank you for submitting your manuscript to PLOS ONE. After careful consideration, we feel that it has merit but does not fully meet PLOS ONE’s publication criteria as it currently stands. Therefore, we invite you to submit a revised version of the manuscript that addresses the points raised during the review process.

The article requires some improvements in the abstract and text as suggested by the reviewers.

We look forward to receiving your revised manuscript.

Kind regards,

Aldrin V. Gomes, Ph.D.

Academic Editor

PLOS ONE

Journal Requirements:

Reviewers' comments:

Reviewer's Responses to Questions

**Comments to the Author**

1. If the authors have adequately addressed your comments raised in a previous round of review and you feel that this manuscript is now acceptable for publication, you may indicate that here to bypass the “Comments to the Author” section, enter your conflict of interest statement in the “Confidential to Editor” section, and submit your "Accept" recommendation.

Reviewer #1: (No Response)

Reviewer #2: (No Response)

2. Is the manuscript technically sound, and do the data support the conclusions?

Reviewer #1: Yes

Reviewer #2: Yes

3. Has the statistical analysis been performed appropriately and rigorously? 

Reviewer #1: Yes

Reviewer #2: Yes

4. Have the authors made all data underlying the findings in their manuscript fully available?

Reviewer #1: Yes

Reviewer #2: Yes

5. Is the manuscript presented in an intelligible fashion and written in standard English?

Reviewer #1: Yes

Reviewer #2: Yes

6. Review Comments to the Author

Reviewer #1: This is an improved version of the detailed research article that unraveled the role of ferroptosis and its inhibition in diabetic kidney disease(DKD) using in-vitro HK-2 cells and male diabetic mice model. The experiments in this manuscript were well executed and the rationale behind the experiments support the hypothesis and final conclusion, Although interesting, informative, and thorough, I would suggest a major revision of the Introduction section as it happens to contain a lot of typographical and grammatical errors that cannot be overlooked.

Due to the standard of this journal, I would recommend this manuscript to be accepted only after if the author are able to fix the grammatical errors in the manuscript.

Some examples of concerns that needs to be rectified but not limited to:

1. Abstract, line 4 and 5: Ferroptosis is a novelty the term linked to lipid hydroperoxidation, and it takes an important part in the pathogenesis of DKD.

Suggestion: It should read "Ferroptosis is a novelty term linked to lipid hydroperoxidation, and it plays an important part/role in the pathogenesis of DKD."

2. "DN models" were mentioned in the manuscript multiple times but I believe it should be DM. Please clarify the meaning of DN.

3. Introduction, line 28: Ferroptosis is a novel nonapoptotic cell death that is discovered iron-dependent form in 2012, induced by a redox imbalance redox homeostasis, which is characterized by detoxify lipid oxidation product and free radicals [11].

4. "These suggesting that by inhibiting ferroptosis in injury PTECs may become a new treatment for kidney"

Suggestion: It should be suggest; ... injured PTECs

5. "COX2 as an cyclooxygenase super-family protein that is an inducible enzyme at low levels in most tissues and

significantly increased in the proximal tubule of the injured kidney [25,26]"

Suggestion: This statement needs to be changed/reworded

6. Results, 3.1, line 22: "Simultaneously, the data showed that the end products of lipid peroxidation, Micro Malondialdehyde Assay Kit was used to assess the level of Malondialdehyde (MDA), the production of MDA was increased by 50% (Fig 1I) compared with the Ctrl group.

Suggestion: This sentence is confusing and would be better rewritten in a succinct manner.

7. "We hypothesized that RSL3 was applied to cause ferroptosis in HK-2 cells in order to further explore the role of COX2 in ferroptosis." This word hypothesized isn't needed in this and might be better in a version similar to "We utilized RSL3 to cause ferroptosis in HK-2 cells in order to further explore the role of COX2 in ferroptosis"

8. Furthermore, the viability of cells treating with RSL-3 was assessed by CCK8.

Suggestion: It should read the viability of cells treated with with RSL-3 was assessed by CCK8.

9. Moreover, the lipid peroxidation product accumulation,MDA(Fig 6K) and high the concentrations of iron (Fig 6L) under ferroptosis conditions were also attenuated after Fer-1 or aspirin treatment.

The word "high the concentrations of iron" is confusing in this sentence.

Please endeavor to check for mistakes in order to allow better comprehension of this article and captivate the readers.

Reviewer #2: The author answered most of the questions, but the author didn't address the 4-HNE level in high glucose incubated HK2 cells.

7. PLOS authors have the option to publish the peer review history of their article (what does this mean?). If published, this will include your full peer review and any attached files.

Reviewer #1: No

Reviewer #2: No

---

## [Author Response · Author response to Decision Letter 1]

25 Oct 2022

Reviewer #1: This article contain some typographical and grammatical errors that cannot be overlooked.

Thank you for your kind suggestion.We have described these sentence in clearer manner in the revised manuscript, as follows:

1. Abstract, line 4 and 5: Ferroptosis is a novelty the term linked to lipid hydroperoxidation, and it takes an important part in the pathogenesis of DKD.

Reply: Ferroptosis is a novel term linked to lipid hydroperoxidation, and it plays an important role in the pathogenesis of DKD. 

2."DN models" were mentioned in the manuscript multiple times but I believe it should be DM. Please clarify the meaning of DN.

Reply:Thank you for your nice advice.We have removed the DN models from the article.

3. Introduction, line 28: Ferroptosis is a novel nonapoptotic cell death that is discovered iron-dependent form in 2012, induced by a redox imbalance redox homeostasis, which is characterized by detoxify lipid oxidation product and free radicals [11].

Reply:Ferroptosis is a novel nonapoptotic cell death that was discovered to be iron-dependent in 2012 and induced by redox imbalance redox homeostasis, which is characterized by detoxification of lipid oxidation products and free radicals [11].

4. "These suggesting that by inhibiting ferroptosis in injury PTECs may become a new treatment for kidney"

Reply:These results suggest that inhibiting ferroptosis in injured PTECs may become a new treatment for kidney diseases.

5. "COX2 as an cyclooxygenase super-family protein that is an inducible enzyme at low levels in most tissues and significantly increased in the proximal tubule of the injured kidney [25,26]"

Reply:COX2 is a cyclooxygenase superfamily protein that is an inducible enzyme at low levels in most tissues and is significantly increased in the proximal tubule of the injured kidney [25,26]. 

6. Results, 3.1, line 22: "Simultaneously, the data showed that the end products of lipid peroxidation, Micro Malondialdehyde Assay Kit was used to assess the level of Malondialdehyde (MDA), the production of MDA was increased by 50% (Fig 1I) compared with the Ctrl group.

Reply:Simultaneously, the data showed the end products of lipid peroxidation; a micro malondialdehyde assay kit was used to assess the level of malondialdehyde (MDA), and the production of MDA was increased by 50% (Fig. 1I) compared with that in the Ctrl group.

7."We hypothesized that RSL3 was applied to cause ferroptosis in HK-2 cells in order to further explore the role of COX2 in ferroptosis."

Reply:We utilized RSL3 to cause ferroptosis in HK-2 cells in order to further explore the role of COX2 in ferroptosis.

8. Furthermore, the viability of cells treating with RSL-3 was assessed by CCK8.

Reply:Furthermore, the viability of cells treated with RSL-3 was assessed by CCK8 assay.

9. Moreover, the lipid peroxidation product accumulation,MDA(Fig 6K) and high the concentrations of iron (Fig 6L) under ferroptosis conditions were also attenuated after Fer-1 or aspirin treatment.

Reply:Moreover, lipid peroxidation product accumulation, MDA (Fig. 6K) and concentrations of iron (Fig. 6L) under ferroptosis conditions were also attenuated after Fer-1 or aspirin treatment. 

Reviewer #2: The author answered most of the questions, but the author didn't address the 4-HNE level in high glucose incubated HK2 cells.

Thank you for your kind suggestion. The products of lipid peroxidation involve reactive aldehydes (e.g. 4-hydroxynonenal (4HNE) and malondialdehyde (MDA)), which increase during ferroptosis . In this study, we have added experiments on 4-HNE levels of the cells.The products of lipid peroxidation, 4-HNE(Fig. 1J) was dramatically elevated in in HG group cells.Normally, untreated control cells contained only a minimal amount of 4-HNE. The cells treated with high glucose showed the maximal level compared to NC group cells. Interestingly, aspirin treatment markedly attenuated the level compared to that in HG group cells (Fig. 4J). Moreover,the higher levels of 4-HNE were also observed in high glucose-treated HK-2 cells, and this decrease was ameliorated in high glucose-cultured cells transfected with COX2-siRNA (Fig. 5J). The lipid peroxidation product accumulation, 4-HNE(Fig.6L) under ferroptosis induced by cells cultured in 100 nm RSL-3 medium was also attenuated after Fer-1 or aspirin treatment.Simultaneously, pretreatment with transfection of COX2-siRNA reduced the concentrations of 4-HNE（Fig.7K）, which implied that transfection of COX2-siRNA could improve the states of lipid peroxidation induced by cells cultured in 100 nm RSL-3 medium.

---

## [Decision Letter · Decision Letter 2]

29 Nov 2022

A spirin Mediates Protection from Diabetic Kidney Disease by Inducing ferroptosis Inhibition

PONE-D-21-40203R2

Dear Dr. Wu,

We’re pleased to inform you that your manuscript has been judged scientifically suitable for publication and will be formally accepted for publication once it meets all outstanding technical requirements.

Kind regards,

Aldrin V. Gomes, Ph.D.

Academic Editor

PLOS ONE

Additional Editor Comments (optional):

Reviewers' comments:

Reviewer's Responses to Questions

**Comments to the Author**

1. If the authors have adequately addressed your comments raised in a previous round of review and you feel that this manuscript is now acceptable for publication, you may indicate that here to bypass the “Comments to the Author” section, enter your conflict of interest statement in the “Confidential to Editor” section, and submit your "Accept" recommendation.

Reviewer #1: (No Response)

Reviewer #2: All comments have been addressed

2. Is the manuscript technically sound, and do the data support the conclusions?

Reviewer #1: Yes

Reviewer #2: Yes

3. Has the statistical analysis been performed appropriately and rigorously? 

Reviewer #1: Yes

Reviewer #2: Yes

4. Have the authors made all data underlying the findings in their manuscript fully available?

Reviewer #1: Yes

Reviewer #2: Yes

5. Is the manuscript presented in an intelligible fashion and written in standard English?

Reviewer #1: Yes

Reviewer #2: Yes

6. Review Comments to the Author

Reviewer #1: This is an improved revision of the manuscript. However, minor typographical error I picked up was in results section

3.1:Another products of lipid peroxidation, 4-HNE(Fig. 1J) was dramatically "elevated in in" HG group cells. It should be "elevated in". Another minor but optional recommendation was the NC group abbreviation used in the mouse model during this study. I'm aware its the control group but it might be better if NC meaning is written in the full form at least once to inform the readers what that group is exactly.

Reviewer #2: (No Response)

7. PLOS authors have the option to publish the peer review history of their article (what does this mean?). If published, this will include your full peer review and any attached files.

Reviewer #1: No

Reviewer #2: No

---

## [Editor Report · Acceptance letter]

4 Dec 2022

PONE-D-21-40203R2 

Aspirin Mediates Protection from Diabetic Kidney Disease by Inducing ferroptosis Inhibition 

Dear Dr. Wu:

I'm pleased to inform you that your manuscript has been deemed suitable for publication in PLOS ONE. Congratulations! Your manuscript is now with our production department. 

Kind regards, 

on behalf of

Dr. Aldrin V. Gomes 

Academic Editor

PLOS ONE